# Robust Sparse Regression with Non-Isotropic Designs

**Chih-Hung Liu**
Department of Electrical Engineering
National Taiwan University
chliu@ntu.edtw

**Gleb Novikov**
Lucerne School of Computer Science and Information Technology
gleb.novikov@hslu.ch

## Abstract

We develop a technique to design efficiently computable estimators for sparse linear regression in the simultaneous presence of two adversaries: oblivious and adaptive. Consider the model $y^* = X^*\beta^* + \eta$ where $X^*$ is an $n \times d$ random design matrix, $\beta^* \in \mathbb{R}^d$ is a $k$-sparse vector, and the noise $\eta$ is independent of $X^*$ and chosen by the *oblivious adversary*. Apart from the independence of $X^*$, we only require a small fraction entries of $\eta$ to have magnitude at most 1. The *adaptive adversary* is allowed to arbitrarily corrupt an $\varepsilon$-fraction of the samples $(X_1^*, y_1^*), \ldots, (X_n^*, y_n^*)$. Given the $\varepsilon$-corrupted samples $(X_1, y_1), \ldots, (X_n, y_n)$, the goal is to estimate $\beta^*$. We assume that the rows of $X^*$ are iid samples from some $d$-dimensional distribution $\mathcal{D}$ with zero mean and (unknown) covariance matrix $\Sigma$ with bounded condition number.

We design several robust algorithms that outperform the state of the art even in the special case of Gaussian noise $\eta \sim N(0,1)^n$. In particular, we provide a polynomial-time algorithm that with high probability recovers $\beta^*$ up to error $O(\sqrt{\varepsilon})$ as long as $n \geqslant \tilde{O}(k^2/\varepsilon)$, only assuming some bounds on the third and the fourth moments of $\mathcal{D}$. In addition, prior to this work, even in the special case of Gaussian design $\mathcal{D} = N(0, \Sigma)$ and noise $\eta \sim N(0,1)$, no polynomial time algorithm was known to achieve error $o(\sqrt{\varepsilon})$ in the sparse setting $n < d^2$. We show that under some assumptions on the fourth and the eighth moments of $\mathcal{D}$, there is a polynomial-time algorithm that achieves error $o(\sqrt{\varepsilon})$ as long as $n \geqslant \tilde{O}(k^4/\varepsilon^3)$. For Gaussian distribution $\mathcal{D} = N(0, \Sigma)$, this algorithm achieves error $O(\varepsilon^{3/4})$. Moreover, our algorithm achieves error $o(\sqrt{\varepsilon})$ for all log-concave distributions if $\varepsilon \leqslant 1/\mathrm{polylog(d)}$.

Our algorithms are based on the filtering of the covariates that uses sum-of-squares relaxations, and weighted Huber loss minimization with $\ell_1$ regularizer. We provide a novel analysis of weighted penalized Huber loss that is suitable for heavy-tailed designs in the presence of two adversaries. Furthermore, we complement our algorithmic results with Statistical Query lower bounds, providing evidence that our estimators are likely to have nearly optimal sample complexity.

## 1   Introduction

Linear regression is the fundamental task in statistics, with many applications in data science and machine learning. In ordinary (non-sparse) linear regression, we are given observations $y_1^*, \ldots, y_n^*$ and $X_1^*, \ldots, X_n^* \in \mathbb{R}^d$ such that $y_i^* = \langle X_i^*, \beta^* \rangle + \eta_i$ for some $\beta^* \in \mathbb{R}^d$ and some noise $\eta \in \mathbb{R}^n$, and the goal

38th Conference on Neural Information Processing Systems (NeurIPS 2024).

is to estimate $\beta^*$. If $\eta$ is independent of $X^*$ and has iid Gaussian entries $\eta_i \sim N(0, 1)$, the classical least squares estimator $\hat{\beta}$ with high probability achieves the *prediction error* $\frac{1}{\sqrt{n}}\|X^*(\hat{\beta}-\beta^*)\| \leqslant O\left(\sqrt{d/n}\right)$. Note that if $d/n \to 0$, the error is vanishing.

Despite the huge dimensions of modern data, many practical applications only depend on a small part of the dimensions of data, thus motivating *sparse* regression, where only $k \ll d$ explanatory variables are actually important (i.e., $\beta^*$ is $k$-sparse). In this case we want the error to be small even if we only have $n \ll d$ samples. In this case, there exists an estimator that achieves prediction error $O\left(\sqrt{k\log(d)/n}\right)$ (for $\eta \sim N(0, 1)^n$). However, this estimator requires exponential computation time. Moreover, under a standard assumption from computational complexity theory (**NP** $\not\subset$ **P/poly**), estimators that can be computed in polynomial time require an assumption on $X^*$ called a *restricted eigenvalue condition* in order to achieve error $O\left(\sqrt{k\log(d)/n}\right)$ (see [ZWJ14] for more details). One efficiently computable estimator that achieves error $O\left(\sqrt{k\log(d)/n}\right)$ under the restricted eigenvalue condition is Lasso, that is, a minimizer of the quadratic loss with $\ell_1$ regularizer. In particular, the restricted eigenvalue condition is satisfied for $X^*$ with rows $X_i^* \overset{\text{iid}}{\sim} N(0, \Sigma)$, where $\Sigma$ has condition number $O(1)$, as long as $n \gtrsim k \log d$ (with high probability).

Further we assume that the designs have iid random rows, and the condition number of the covariance matrix is bounded by some constant. In addition, for random designs, we use the *standard* error $\|\Sigma^{1/2}(\hat{\beta} - \beta^*)\|$. Note that when the number of samples is large enough, this error is very close to $\frac{1}{\sqrt{n}}\|X^*(\hat{\beta} - \beta^*)\|$.

Recently, there was an extensive interest in the linear regression with the presence of adversarially chosen outliers. Under the assumption $X_i^* \overset{\text{iid}}{\sim} N(0, \Sigma)$, the line of works [TJSO14, BJKK17, SBRJ19, dNS21, dLN+21] studied the case when the noise $\eta$ is unbounded and chosen by an *oblivious* adversary, i.e., when $\eta$ is an arbitrary vector independent of $X^*$. As was shown in [dLN+21], in this case, it is possible to achieve the same error (up to a constant factor) as for $\eta \sim N(0, 1)^n$ if we only assume that $\Omega(1)$ fraction of the entries of $\eta$ have magnitude at most 1. They analyzed the *Huber loss* estimator with $\ell_1$ regularizer.

Another line of works [BJK15, DT19, MNW22, Tho23] assumed that $\eta$ has iid random entries that satisfy some assumptions on the moments, but an adversarially chosen $\varepsilon$-fraction of $y_1^*, \dots, y_n^*$ is replaced by arbitrary values by an *adaptive adversary* that can observe $X^*$, $\beta^*$ and $\eta$ (so the corruptions can depend on them). [Tho23] showed that for $X^*$ with iid sub-Gaussian rows and $\eta$ with iid sub-Gaussian entries with unit variance, Huber loss estimator with $\ell_1$ regularizer achieves an error of $O\left(\sqrt{k\log(d)/n} + \varepsilon\log(1/\varepsilon)\right)$ with high probability. Note that the second term depends on $\varepsilon$, but not on $n$; hence, even if we take more samples, this term does not decrease (if $\varepsilon$ remains the same). It is inherent: in the presence of the adaptive adversarial outliers, even for $X_i^* \overset{\text{iid}}{\sim} N(0, \text{Id})$ and $\eta \sim N(0, 1)^n$, the information theoretically optimal error is $\Omega\left(\sqrt{k\log(d)/n} + \varepsilon\right)$, so independently of the number of samples, it is $\Omega(\varepsilon)$. In the algorithmic high-dimensional robust statistics, we are interested in estimators that are computable in time $\text{poly}(d)$. There is evidence that it is unlikely that $\text{poly}(d)$-time computable estimators can achieve error $O(\varepsilon)$ [DKS17]. Furthermore, for other design distributions the optimal error can be different.

Hence the natural questions to ask are : Given an error bound $f(\varepsilon)$, does there exist a $\text{poly}(d)$-time computable estimator that achieves error at most $f(\varepsilon)$ with high probability? If possible, what is the smallest number of samples $n$ that is enough to achieve error $f(\varepsilon)$ in time $\text{poly}(d)$? In the rest of this section, we write error bounds in terms of $\varepsilon$ and mention the number of samples that is required to achieve this error. In addition, we focus on the results for the high dimensional regime, where $f(\varepsilon)$ does not depend polynomially on $k$ or $d$.

Another line of works [BDLS17, LSLC20, PJL20, Sas22, SF23] considered the case when the adaptive adversary is allowed to corrupt $\varepsilon$-fraction of all observed data, i.e. not only $y_1^*, \dots, y_n^*$, but also $X_1^*, \dots, X_n^*$, while the noise $\eta$ is assumed to have iid random entries that satisfy some concentration assumptions. For simplicity, to fix the scale of the noise, we formulate their results

assuming that $\eta \sim N(0, 1)^n$. In non-sparse settings, [PJL20] showed that in the case of identity covariance sub-Gaussian designs, Huber loss minimization after a proper *filtering* of $X^*$ achieves error $\tilde{O}(\varepsilon)$ with $n \gtrsim d/\varepsilon^2$ samples. Informally speaking, filtering removes the samples $X_i^*$ that look corrupted, and if the distribution of the design is nice enough, then after filtering we can work with $(X^*, y^*)$ just like in the case when only $y^*$ is corrupted. For unknown covariance they showed a bound $O(\sqrt{\varepsilon})$ for a large class of distributions of the design. If $X_i^* \overset{\text{iid}}{\sim} N(0, \Sigma)$ for unknown $\Sigma$, one can use $n \geqslant \tilde{O}(d^2/\varepsilon^2)$ samples to robustly estimate the covariance, and achieve nearly optimal error $\tilde{O}(\varepsilon)$ in the case (see [DKS19] for more details).

In the sparse setting, there is likely an information-computation gap for the sample complexity of this problem, even in the case of the isotropic Gaussian design $X_i^* \overset{\text{iid}}{\sim} N(0, \text{Id})$. While it is information-theoretically possible to achieve optimal error $O(\varepsilon)$ with $n \geqslant \tilde{O}(k/\varepsilon^2)$ samples, achieving *any* error $o(1)$ is likely to be not possible for poly($d$)-time computable estimators if $n \ll k^2$. Formal evidence for this conjecture include reductions from some version of the Planted Clique problem [BB20], as well as a Statistical Query lower bound (Proposition 1.10). For $n \geqslant \tilde{O}(k^2/\varepsilon^2)$, several algorithmic results are known to achieve error $\tilde{O}(\varepsilon)$, in particular, [BDLS17, LSLC20], and [SF23] for more general isotropic sub-Gaussian designs. Similarly to the approach of [PJL20], [SF23] used ($\ell_1$-penalized) Huber minimization after filtering $X^*$.

The non-isotropic case (when $\Sigma \neq \text{Id}$ is unknown) is more challenging. [SF23] showed that for sub-Gaussian designs it is possible to achieve error $O(\sqrt{\varepsilon})$ with $n \geqslant \tilde{O}(k^2)$ samples. [Sas22] showed that $O(\sqrt{\varepsilon})$ error with $n \geqslant \tilde{O}(k^2 + \|\beta^*\|_1^4/k^2)$ samples can be achieved under some assumptions on the fourth and the eighth moments of the design distribution. While this result works for a large class of designs, the clear disadvantage is that the sample complexity depends polynomially on the norm of $\beta^*$. For example, if all nonzero entries of $\beta^*$ have the same magnitude and $\|\beta^*\| = \sqrt{d}$, then the sample complexity is $n > d^2$, which is not suitable in the sparse regime.

Prior to this work, no poly($d$)-time computable estimator that could achieve error $o(\sqrt{\varepsilon})$ with unknown $\Sigma$ was known, even in the case of Gaussian designs $X_i^* \overset{\text{iid}}{\sim} N(0, \Sigma)$ and the Gaussian noise $\eta \sim N(0, 1)^n$ (apart from the non-sparse setting, where such estimators require $n > d^2$).

## 1.1 Results

We present two main results, both of them follow from a more general statement; see Theorem B.3. Before formally stating the results, we define the model as follows.

**Definition 1.1** (Robust Sparse Regression with 2 Adversaries). Let $n, d, k \in \mathbb{N}$ such that $k \leqslant d$, $\sigma > 0$, and $\varepsilon \in (0, 1)$ is smaller than some sufficiently small absolute constant. Let $\mathcal{D}$ be a probability distribution in $\mathbb{R}^d$ with mean 0 and covariance $\Sigma$. Let $y^* = X^*\beta^* + \eta$, where $X^*$ is an $n \times d$ random matrix with rows $X_i^* \overset{\text{iid}}{\sim} \mathcal{D}$, $\beta^* \in \mathbb{R}^d$ is $k$-sparse, $\eta \in \mathbb{R}^n$ is independent of $X^*$ and has at least $0.01 \cdot n$ entries bounded by $\sigma$ in absolute value[1]. We denote by $\kappa(\Sigma)$ the condition number of $\Sigma$.

An instance of our model is a pair $(X, y)$, where $X \in \mathbb{R}^{n \times d}$ is a matrix and $y \in \mathbb{R}^n$ is a vector such that there exists a set $S_{\text{good}} \subseteq [n]$ of size at least $(1 - \varepsilon)n$ such that for all $i \in S_{\text{good}}$, $X_i = X_i^*$ and $y_i = y_i^*$.

Note that random noise models studied in prior works are captured by our model in Definition 1.1. For example, if $\eta$ has iid entries that satisfy $\mathbb{E}|\eta_i| \leqslant \sigma/2$, by Markov's inequality, $|\eta_i| \leqslant \sigma$ with probability at least $1/2$, and with overwhelming probability, at least $0.01 \cdot n$ entries of $\eta$ are bounded by $\sigma$ in absolute value. In addition, Cauchy noise (that does not have the first moment) with location parameter 0 and scale $\sigma$ also satisfies these assumptions, as well as other heavy-tailed distributions studied in literature (with appropriate scale parameter $\sigma$).

---

[1]Our result also works for more general model, where we require $\alpha n$ entries to be bounded by $\sigma$ for some $\alpha \gtrsim \varepsilon$. The error bound in this case also depends on $\alpha$.

We formulate our results assuming that the condition number of the covariance is bounded by some constant: $\kappa(\Sigma) \leqslant O(1)$. In the most general formulation (Theorem B.3), we show the dependence[2] of the number of samples and the error on $\kappa(\Sigma)$.

### 1.1.1 Robust regression with heavy-tailed designs

We use the following notion of boundness of the moments of $\mathcal{D}$:

**Definition 1.2.** Let $M > 0$, $t \geqslant 2$ and $d \in \mathbb{N}$. We say that a probability distribution $\mathcal{D}$ in $\mathbb{R}^d$ with zero mean and covariance $\Sigma$ has *M-bounded t-th moment*, if for all $u \in \mathbb{R}^d$

$$\Big( \mathop{\mathbb{E}}_{x \sim \mathcal{D}} |\langle x, u \rangle|^t \Big)^{1/t} \leqslant M \cdot \sqrt{\|\Sigma\|} \cdot \|u\| \,.$$

Note that an arbitrary linear transformation of an *isotropic* distribution with $M$-bounded $t$-th moment also has $M$-bounded $t$-th moment. Also note that if $t' \leqslant t$ and a distribution $\mathcal{D}$ has $M$-bounded $t$-th moment, then the $t'$-th moment of $\mathcal{D}$ is also $M$-bounded. In particular, $M$ cannot be smaller than 1, since the second moment cannot be $M$-bounded for $M < 1$. In addition, we will need the following (weaker) notion of the boundness of moments:

**Definition 1.3.** Let $\nu > 0$, $t \geqslant 2$ and $d \in \mathbb{N}$. We say that a probability distribution $\mathcal{D}$ in $\mathbb{R}^d$ with zero mean and covariance $\Sigma$ has *entrywise $\nu$-bounded t-th moment*, if

$$\max_{j \in [d]} \mathop{\mathbb{E}}_{x \sim \mathcal{D}} |x_j|^t \leqslant \nu^t \cdot \|\Sigma\|^{t/2} \,.$$

If a distribution has $M$-bounded $t$-th moment, then it also has entrywise $M$-bounded $t$-th moment, but the converse might not be true for some distributions. Now we are ready to state our first result.

**Theorem 1.4.** *Let $n, d, k, X, y, \varepsilon, \mathcal{D}, \Sigma, \sigma, \beta^*$ be as in Definition 1.1. Suppose that $\kappa(\Sigma) \leqslant O(1)$ and that for some $1 \leqslant M \leqslant O(1)$ and $1 \leqslant \nu \leqslant O(1)$, $\mathcal{D}$ has M-bounded 3-rd moment and entrywise $\nu$-bounded 4-th moment. There exists an algorithm that, given $X, y, k, \varepsilon, \sigma$, in time $(n + d)^{O(1)}$ outputs $\hat{\beta} \in \mathbb{R}^d$ such that if $n \gtrsim k^2 \log(d)/\varepsilon$, then with probability at least $1 - d^{-10}$,*

$$\|\Sigma^{1/2}(\hat{\beta} - \beta^*)\| \leqslant O(\sigma \cdot \sqrt{\varepsilon}) \,.$$

Let us compare Theorem 1.4 with the state of the art. For heavy-tailed designs, prior to this work, the best estimator was [Sas22]. That estimator also achieves error $O(\sigma\sqrt{\varepsilon})$, but its sample complexity depends polynomially on the norm of $\beta^*$, while our sample complexity does not depend on it. In addition, they require the distribution to have bounded 4-th moment (as opposed to our 3-rd moment assumption), and bounded entrywise 8-th moment (as opposed to our entrywise 4-th moment assumption). Finally, our noise assumption is weaker than theirs since they required the entries of $\eta$ to be iid random variables such that $\mathbb{E}|\eta_i| \leqslant \sigma'$ for some $\sigma' > 0$ known to the algorithm designer; as we mentioned after Definition 1.1, it is a special case of the oblivious noise with $\sigma = 2\sigma'$.

Let us also discuss our assumptions and possibilities of an improvement of our result. The third moment assumption can be relaxed, more precisely, it is enough to require the $t$-th moment to be bounded, where $t$ is an arbitrary constant greater than 2, and in this case the sample complexity is increased by a constant factor[3]; see Theorem B.3 for more details. The entrywise fourth moment assumption is not improvable with our techniques, that is, we get worse dependence on $k$ if we relax it to, say, the third moment assumption.

The dependence of $n$ on $\varepsilon$ is not improvable with our techniques[4]. The dependence of the error on $\sigma$ is optimal. The dependence of $n$ on $k$ and the error on $\sqrt{\varepsilon}$ is likely to be (nearly) optimal: Statistical Query lower bounds (Proposition 1.10 and Proposition 1.11) provide evidence that for $\sigma = \Theta(1)$, it is unlikely that polynomial-time algorithms can achieve error $o(1)$ if $n \ll k^2$, or error $o(\sqrt{\varepsilon})$ if $n \ll k^4$.

*Remark* 1.5. Our results also imply bounds on other types of error studied in literature. In particular, observe that $\|\hat{\beta} - \beta^*\| \leqslant \|\Sigma^{1/2}(\hat{\beta} - \beta^*)\|/\sqrt{\lambda_{\min}(\Sigma)}$, where $\lambda_{\min}(\Sigma)$ is the minimal eigenvalue of $\Sigma$.

---

[2]We did not aim to optimize this dependence.

[3]This factor depends on $M$ and $\kappa(\Sigma)$, as well as on $t$. In particular, it goes to infinity when $t \to 2$.

[4]Some dependence of $n$ on $\varepsilon$ is inherent, but potentially our dependence could be suboptimal. For sub-exponential distributions it is possible to get better dependence, see Remark 1.9 and Appendix H.

In addition, our estimator also satisfies $\|\hat{\beta} - \beta^*\|_1 \leqslant O(\|\Sigma^{1/2}(\hat{\beta} - \beta^*)\| \cdot \sqrt{k/\lambda_{\min}(\Sigma)})$. The same is also true for our estimator from Theorem 1.7 below. These relations between different types of errors are standard for sparse regression, and they are not improvable.

### 1.1.2 Beyond $\sqrt{\varepsilon}$ error

Prior to this work, no polynomial-time algorithm for (non-isotropic) robust sparse regression was known to achieve error $o(\sigma\sqrt{\varepsilon})$, even for Gaussian designs $X_i^* \overset{\text{iid}}{\sim} N(0, \Sigma)$ and Gaussian $\eta \sim N(0, \sigma)^n$. In this section we show that for a large class of designs, it is possible to achieve error $o(\sigma\sqrt{\varepsilon})$ in polynomial time, even when $\eta$ is chosen by an oblivious adversary. For our second result, we require not only some bounds on the moments of $\mathcal{D}$, but also their certifiability in the *sum-of-squares proof system*:

**Definition 1.6.** Let $M > 0$ and let $\ell \geqslant 4$ be an even number. We say that a probability distribution $\mathcal{D}$ in $\mathbb{R}^d$ with zero mean and covariance $\Sigma$ has $\ell$-*certifiably $M$-bounded 4-th moment*, if there exist polynomials $h_1, \ldots, h_m \in \mathbb{R}[u_1, \ldots, u_d]$ of degree at most $\ell/2$ such that

$$\mathbb{E}_{x \sim \mathcal{D}} \langle x, u \rangle^4 + \sum_{i=1}^{m} h_i^2(u) = M^4 \cdot \|\Sigma\|^2 \cdot \|u\|^4 .$$

Definition 1.6 with arbitrary $\ell$ implies Definition 1.2 (with the same $M$). Under standard complexity-theoretic assumptions, there exist distributions with bounded moments that are not $\ell$-certifiably bounded even for very large $\ell$ [HL19]. Note that similarly to Definition 1.2, an arbitrary linear transformation of an isotropic distribution with $\ell$-certifiably $M$-bounded 4-th moment also has $\ell$-certifiably $M$-bounded 4-th moment.

Distributions with certifiably bounded moments are very important in algorithmic robust statistics. They were extensively studied in literature, e.g. [KS17a, KS17b, HL18, HL19, DKK$^+$22].

Now we can state our second result.

**Theorem 1.7.** *Let $n, d, k, X, y, \varepsilon, \mathcal{D}, \Sigma, \sigma, \beta^*$ be as in Definition 1.1. Suppose that $\kappa(\Sigma) \leqslant O(1)$, and that for some $M \geqslant 1$, some even number $\ell \geqslant 4$, and $1 \leqslant \nu \leqslant O(1)$, $\mathcal{D}$ has $\ell$-certifiably $M$-bounded 4-th moment and entrywise $\nu$-bounded 8-th moment. There exists an algorithm that, given $X, y, k, \varepsilon, \sigma, M, \ell$, in time $(n + d)^{O(\ell)}$ outputs $\hat{\beta} \in \mathbb{R}^d$ such that if $n \gtrsim M^4 \cdot k^4 \log(d)/\varepsilon^3$, then with probability at least $1 - d^{-10}$,*

$$\|\Sigma^{1/2}(\hat{\beta} - \beta^*)\| \leqslant O(M \cdot \sigma \cdot \varepsilon^{3/4}) .$$

In particular, in the regime $M \leqslant O(1)$, as long as $n \geqslant \tilde{O}(k^4/\varepsilon^3)$, the algorithm recovers $\beta^*$ from $(X, y)$ up to error $O(\sigma\varepsilon^{3/4})$ (with high probability). If $\ell \leqslant O(1)$, the algorithm runs in polynomial time. Note that in this theorem we do not assume that $M$ is constant as opposed to Theorem 1.4 since for some natural classes of distributions, only some bounds on $M$ that depend on $d$ are known.

The natural question is what distributions have certifiably bounded fourth moment with $\ell \leqslant O(1)$. First, these are products of one-dimensional distributions with $M$-bounded fourth moment, and their linear transformations (with $\ell = 4$). Hence, linear transformations of products of one-dimensional distributions with $O(1)$-bounded 8-th moment satisfy the assumptions of the theorem with $M \leqslant O(1)$ and $\ell = 4$. Note that such distributions might not even have a 9-th moment. This class also includes Gaussian distributions (since they are linear transformations of the $N(0, 1)^d$ and $N(0, 1)$ has $O(1)$-bounded 8-th moment).

Another important class is the distributions that satisfy *Poincaré inequality*. Concretely, these distributions, for some $C_P \geqslant 1$, satisfy $\text{Var}_{x \sim \mathcal{D}} g(x) \leqslant C_P^2 \cdot \|\Sigma\| \cdot \mathbb{E}_{x \sim \mathcal{D}}\|\nabla g(x)\|_2^2$ for all continuously differentiable functions $g : \mathbb{R}^d \to \mathbb{R}$. [KS17a] showed that such distributions have 4-certifiably $O(C_P)$-bounded fourth moment. We will not further discuss Poincaré inequality, and focus on the known results on the classes of distributions satisfy this inequality.

The Kannan-Lovász-Simonovits (KLS) conjecture from convex geometry says that $C_P$ is bounded by some universal constant for *all* log-concave distributions. Recall that a distribution $\mathcal{D}$ is called log-concave if for some convex function $V : \mathbb{R}^d \to \mathbb{R}$, the density of $\mathcal{D}$ is proportional to $e^{-V(x)}$.

Apart from the Gaussian distribution, examples include uniform distributions over convex bodies, the Wishart distribution and the Dirichlet distribution ([Pré71], see also [KBJ00] for further examples). In recent years there has been a big progrees towards the proof of the KLS conjecture. [Che21] showed that $C_P \leqslant d^{o(1)}$, and since then, the upper bound has been further significantly improved. The best current bound is $C_P \leqslant O(\sqrt{\log d})$ obtained by [Kla23]. This bound implies that for all log-concave distributions whose covariance has bounded condition number, the error of our estimator is $O(\sigma\sqrt{\log d} \cdot \varepsilon^{3/4})$. Hence for $\varepsilon \leqslant o(1/\log^2(d))$ and $\sigma \leqslant O(1)$, the error is $o(\sqrt{\varepsilon})$. Note that if the KLS conjecture is true, the error of our estimator is $O(\sigma\varepsilon^{3/4})$ for all log-concave distributions with $\kappa(\Sigma) \leqslant O(1)$, without any restrictions on $\varepsilon$ (except the standard $\varepsilon \lesssim 1$).

*Remark* 1.8. Theorem 1.7 can be generalized as follows: If the $(2t)$-th moment of $\mathcal{D}$ is $M$-bounded for a constant $t \in \mathbb{N}_{\geqslant 2}$, if this bound can be certified by a constant degree sum-of-squares proof[5], and if $\mathcal{D}$ has entrywise $(4t)$-th $O(1)$-bounded moment, then with high probability, there is a poly$(d)$-time computable estimator that achieves error $O(M\sigma\varepsilon^{1-1/(2t)})$ as long as $n \gtrsim M^4 k^{2t} \log(d)/\varepsilon^{2t-1}$. See Theorem B.3 for more details.

*Remark* 1.9. The dependence of $n$ on $\varepsilon$ can be improved under the assumption that $\mathcal{D}$ is a *sub-exponential* distribution. In particular, all log-concave distributions are sub-exponential. Under this additional assumption, in order to achieve the error $O(\sigma\sqrt{\varepsilon})$, it is enough to take $n \gtrsim k^2 \operatorname{polylog}(d) + k \log(d)/\varepsilon$, and to achieve error $O(M\sigma\varepsilon^{3/4})$, it is enough to take $n \gtrsim k^4 \operatorname{polylog}(d) + k \log(d)/\varepsilon^{3/2}$ samples (assuming, as in Theorem 1.7, that the fourth moment is $M$-certifiably bounded).

### 1.1.3   Lower bounds

We provide *Statistical Query* (SQ) lower bounds by which our estimators likely have optimal sample complexities needed to achieve the errors $O(\sqrt{\varepsilon})$ and $o(\sqrt{\varepsilon})$, even when the design and the noise are Gaussian. SQ lower bounds are usually interpreted as a tradeoff between the time complexity and sample complexity of estimators; see Appendix G and [DKS17] for more details. Our proofs are very similar to prior works [DKS17, DKS19, DKK$^+$22] since as was observed in [DKS19], lower bounds for mean estimation can be used to prove lower bounds for linear regression, and we use the lower bounds for sparse mean estimation from [DKS17, DKK$^+$22].

Let us fix the scale of the noise $\sigma = 1$. The first proposition shows that already for $\Sigma = \mathrm{Id}$, $k^2$ samples are likely to be necessary to achieve error $o(1)$:

**Proposition 1.10** (Informal, see Proposition G.9). *Let $n, d, k, X, y, \varepsilon, \mathcal{D}, \Sigma, \sigma, \beta^*$ be as in Definition 1.1. Suppose that $\mathcal{D} = N(0, \mathrm{Id})$ and $\eta \sim N(0, \tilde{\sigma}^2)^n$, where $0.99 \leqslant \tilde{\sigma} \leqslant 1$. Suppose that $d^{0.01} \leqslant k \leqslant \sqrt{d}$, $\varepsilon \gtrsim \frac{1}{\sqrt{\log d}}$, and $n \leqslant k^{1.99}$. Then for each SQ algorithm A that finds $\hat{\beta}$ such that $\|\beta^* - \hat{\beta}\| \leqslant 10^{-5}$, the simulation of A with n samples has to simulate super-polynomial ($\exp(d^{\Omega(1)})$) number of queries.*

Note that under assumptions of Proposition 1.10, Theorem 1.4 implies that if we take $n \geqslant k^2 \operatorname{polylog}(d)$ samples, the estimator achieves error $O(\sqrt{\varepsilon})$ that is $o(1)$ if $\varepsilon \to 0$ as $d \to \infty$.

The second proposition shows that for $\frac{1}{2} \preceq \Sigma \preceq \mathrm{Id}$, $k^4$ samples are likely to be necessary to achieve error $o(\sqrt{\varepsilon})$:

**Proposition 1.11** (Informal, see Proposition G.10). *Let $n, d, k, X, y, \varepsilon, \mathcal{D}, \Sigma, \sigma, \beta^*$ be as in Definition 1.1. Suppose that $\mathcal{D} = N(0, \Sigma)$ for some $\Sigma$ such that $\frac{1}{2} \preceq \Sigma \preceq \mathrm{Id}$, and $\eta \sim N(0, \tilde{\sigma}^2)^n$, where $0.99 \leqslant \tilde{\sigma} \leqslant 1$. Suppose that $d^{0.01} \leqslant k \leqslant \sqrt{d}$, $\varepsilon \gtrsim \frac{1}{\log d}$, and $n \leqslant k^{3.99}$. Then for each SQ algorithm A that finds $\hat{\beta}$ such that $\|\beta^* - \hat{\beta}\| \leqslant 10^{-5}\sqrt{\varepsilon}$, the simulation of A with n samples has to simulate super-polynomial ($\exp(d^{\Omega(1)})$) number of queries.*

Note that under assumptions of Proposition 1.11, Theorem 1.7 implies that if we take $n \geqslant k^4 \operatorname{polylog}(d)$ samples, the estimator achieves error $O(\varepsilon^{3/4})$ that is $o(\sqrt{\varepsilon})$ if $\varepsilon \to 0$ as $d \to \infty$.

---

[5]See Definition B.2 for formal definition.

## 2 Techniques

Since the problem has multiple aspects, we first illustrate our approach on the simplest example $X_i^* \overset{\text{iid}}{\sim} N(0, \Sigma)$ under the assumption that $0.1 \cdot \text{Id} \preceq \Sigma \preceq 10 \cdot \text{Id}$. Note that already in this case, even for $\eta \sim N(0,1)^n$, our estimator from Theorem 1.7 outperforms the state of the art. In addition, we assume that $\sigma = 1$.

Our estimators are based on preprocessing $X$, and then minimizing $\ell_1$-*penalized Huber loss*. In the Gaussian case, the preprocessing step consists only of *filtering*, while for heavy-tailed designs, an additional *truncation* step is required. The idea of using filtering before minimizing the Huber loss first appeared in [PJL20] for the dense settings, and was applied to sparse settings in [Sas22, SF23]. We will not discuss the filtering method in detail, and rather focus on its outcome: It is a set $\hat{S} \subseteq [n]$ of size at least $(1 - O(\varepsilon))n$ that satisfies some nice properties[6]. Further, we will see what properties we need from $\hat{S}$, and now let us define the Huber loss estimator.

**Definition 2.1.** For $S \subseteq [n]$, the *Huber loss function restricted to $S$* is defined as

$$H_S(\beta) = \frac{1}{n} \sum_{i \in S} h(\langle X_i, \beta \rangle - y_i) \text{ where } h(x_i) = \begin{cases} \frac{1}{2}x_i^2 & \text{if } |x_i| \leqslant 2; \\ 2|x_i| - 2 & \text{otherwise.} \end{cases}$$

For a penalty parameter $\lambda$, the $\ell_1$-penalized Huber loss restricted to $S$ is defined as $L_S(\beta) := H_S(\beta) + \lambda \cdot \|\beta\|_1$. We use the notation $\phi(x)$ for the derivative of $h(x)$. Note that for all $x$, $|\phi(x)| \leqslant 2$.

Our estimator is the minimizer $\hat{\beta}_{\hat{S}}$ of $L_{\hat{S}}(\beta)$, where $\hat{S}$ is the set returned by the filtering algorithm. To investigate the properties of this estimator, it is convenient to work with *elastic balls*. The $k$-elastic ball of radius $r$ is the following set: $\mathcal{E}_k(r) := \{u \in \mathbb{R}^d \mid \|u\| \leqslant r, \|u\|_1 \leqslant \sqrt{k} \cdot r\}$. Note that this ball contains all $k$-sparse vectors with Euclidean norm at most $r$ (as well as some other vectors). Elastic balls are very useful for sparse regression since if the following two properties hold,

1. *Gradient bound:* For all $u \in \mathcal{E}_k(r)$,   $|\langle \nabla H_{\hat{S}}, u \rangle| \lesssim \frac{r}{\sqrt{k}}\|u\|_1 + r\|u\|$,

2. *Strong convexity on the boundary:* For all $u \in \mathcal{E}_k(r)$ such that $\|u\| = r$,

$$H_{\hat{S}}(\beta^* + u) - H_{\hat{S}}(\beta^*) - \langle \nabla H_{\hat{S}}, u \rangle \geqslant \Omega(r^2),$$

then for an appropriate choice of the penalty parameter $\lambda$, then $\|\beta^* - \hat{\beta}_{\hat{S}}\| < r$.[7]

Hence it is enough to show these two properties. In the Gaussian case, the strong convexity property can be proved in exactly the same way as it is done in [dLN+21] for the case of the oblivious adversary, while for heavy-tailed designs it is significantly more challenging. Since we now discuss the Gaussian case, let us focus on the gradient bound. Denote $H_S^*(\beta) = \frac{1}{n} \sum_{i \in S} h(\langle X_i^*, \beta \rangle - y_i^*)$. By triangle inequality,

$$|\langle \nabla H_{\hat{S}}, u \rangle| = |\langle \nabla H_{S_{\text{good}} \cap \hat{S}}^*, u \rangle + \langle \nabla H_{S_{\text{bad}} \cap \hat{S}}, u \rangle|$$

$$\leqslant |\langle \nabla H_{[n]}^*, u \rangle| + |\langle \nabla H_{[n] \setminus (S_{\text{good}} \cap \hat{S})}^*, u \rangle| + |\langle \nabla H_{S_{\text{bad}} \cap \hat{S}}, u \rangle|.$$

Since the first term can be bounded by $\|\nabla H_{[n]}^*\|_\infty \cdot \|u\|_1$, it is enough to show that $\|\nabla H_{[n]}^*\|_\infty \lesssim r/\sqrt{k}$, where $r$ is the error we aim to achieve. Note that $\nabla H_{[n]}^* = \frac{1}{n} \sum_{i=1}^n \phi(\eta_i)\langle X_i^*, u \rangle$ does not depend on the outliers created by the adaptive adversary. The sharp bound on $\|\nabla H_{[n]}^*\|_\infty$ can be derived in exactly the same way as in [dLN+21] (or other prior works): Since $\eta$ and $X^*$ are independent and $|\phi(\eta)| \leqslant 2$, $\nabla H_{[n]}^*$ is a Gaussian vector whose entries have variance $(1/n)$. By standard properties of Gaussian vectors, $\|\nabla H_{[n]}^*\|_\infty \leqslant O(\sqrt{\log(d)/n})$ with high probability.

---

[6]Technically, the filtering we use returns weights of the samples. For simplicity we assume here that the weights are 0 or 1.

[7]For simplicity, we omit some details, e.g. we need to work with $\mathcal{E}_{k'}(r)$ instead of $\mathcal{E}_k(r)$, where $k' \gtrsim k$. See Theorem A.3 for the formal statement. Similar statements appeared in many prior works on sparse regression.

To bound the second and the third term, we can use Cauchy–Schwarz inequality and get $O(\sqrt{\varepsilon})$ dependence on the error (like it is done in prior works on robust sparse regression, for example, [Sas22] or [SF23]), or use Hölder's inequality and get better dependence, but also more challenges since we have to work with higher (empirical) moments of $X^*$ and $X$. Let us use Hölder'sinequality and illustrate how we work with higher moments. Note that both sets $[n] \setminus (S_{\text{good}} \cap \hat{S})$ and $S_{\text{bad}} \cap \hat{S}$ have size at most $O(\varepsilon n)$. Hence the second term can be bounded by

$$O(\varepsilon^{3/4}) \cdot \Big( \sum_{i \in [n] \setminus (S_{\text{good}} \cap \hat{S})} \tfrac{1}{n} \langle X_i^*, u \rangle^4 \Big)^{1/4} \leqslant O(\varepsilon^{3/4}) \cdot \Big( \sum_{i \in [n]} \tfrac{1}{n} \langle X_i^*, u \rangle^4 \Big)^{1/4},$$

while the third term is bounded by

$$O(\varepsilon^{3/4}) \cdot \Big( \sum_{i \in S_{\text{bad}} \cap \hat{S}} \tfrac{1}{n} \langle X_i, u \rangle^4 \Big)^{1/4} \leqslant O(\varepsilon^{3/4}) \cdot \Big( \sum_{i \in \hat{S}} \tfrac{1}{n} \langle X_i, u \rangle^4 \Big)^{1/4}.$$

A careful probabilistic analysis shows that with high probability, for all $r \geqslant 0$ and all $u \in \mathcal{E}_k(r)$, $\sum_{i \in [n]} \tfrac{1}{n} \langle X_i^*, u \rangle^4 \leqslant O(\|u\|^4)$. Hence, our requirement on $\hat{S}$ is that $\sum_{i \in \hat{S}} \tfrac{1}{n} \langle X_i, u \rangle^4 \leqslant O(1)$ for all $u \in \mathcal{E}_k(1)$ (by scaling argument, it is enough to consider $r = 1$). If we find such a set $\hat{S}$, we get the desired bound. Indeed, if $n \gtrsim k \log(d)/\varepsilon^{3/2}$, $\|\nabla H_{[n]}^*\|_\infty \leqslant O(\varepsilon^{3/4}/\sqrt{k})$, and the other terms are bounded by $O(\varepsilon^{3/4})$, implying that $\|\hat{\beta} - \hat{\beta}_{\hat{S}}\| < r = O(\varepsilon^{3/4})$.

Note that such sets of size $(1 - O(\varepsilon))n$ exist since $S_{\text{good}}$ satisfies this property. It is clear how to find such a set inefficiently: we just need to check all candidate sets $S$ and maximize the quartic function $\sum_{i \in S} \langle X_i, u \rangle^4$ over $u \in \mathcal{E}_k(1)$. Furthermore, the by-now standard filtering method allows to avoid checking all the sets: If we can maximize $\sum_{i \in S} \langle X_i, u \rangle^4$ over $u \in \mathcal{E}_k(1)$ efficiently, we can also find the desired set efficiently.

Before explaining how we maximize this function, let us see how prior works [BDLS17, SF23], optimized a simpler quadratic function $\sum_{i \in S} \langle X_i, u \rangle^2$ over $u \in \mathcal{E}_k(1)$. They use the *basic SDP* relaxation for sparse PCA, that is, they optimize the linear function $\sum_{i \in S} \langle X_i X_i^\top, U \rangle$ over $\mathcal{B}_k :=$ $\{U \in \mathbb{R}^{d \times d} \mid U \geq 0, \operatorname{Tr}(U) \leqslant 1, \|U\|_1 \leqslant k\}$. This set has been used in literature for numerous sparse problems since it is a nice (perhaps the best) convex relaxation of the set $\mathcal{S}_k = \{uu^\top \mid u \in \mathbb{R}^d, \|u\| \leqslant 1, \|u\|_0 \leqslant k\}$. Moreover, crucially for sparse regression, it is easy to see that $\mathcal{B}_k$ also contains all matrices $uu^\top$ such that $u \in \mathcal{E}_k(1)$. Hence, one may try to optimize quartic functions by using relaxations of $\mathcal{S}_k = \{u^{\otimes 4} \mid u \in \mathbb{R}^d, \|u\| \leqslant 1, \|u\|_0 \leqslant k\}$. A natural relaxation is the sum-of-squares with *sparsity constraints*. [DKK+22] used these relaxations for sparse mean estimation[8]. They showed that these relaxations provide nice guarantees for distributions with certifiably bounded 4-th moment, assuming that the distribution has sub-exponential tails. Since we now discuss the Gaussian case, the assumption on the tails is satisfied. However, there is no guarantee that these relaxations capture $u^{\otimes 4}$ for all $u \in \mathcal{E}_k(1)$. So, for sparse regression, we need another relaxation.

We use the sum-of-squares relaxations with *elastic constraints*. These constraints ensure that the set of relaxations $\mathcal{P}_k \subset \mathbb{R}^{d^4}$ is guaranteed to contain $u^{\otimes 4}$ for all $u \in \mathcal{E}_k(1)$. We show that if $n \gtrsim \tilde{O}(k^4)$, there is a degree-$O(1)$ sum-of-squares proof from the elastic constraints of the fact that $\frac{1}{n} \sum_{i \in [n]} \langle X_i, u \rangle^4 \leqslant O(1)$. It implies that the relaxation is nice: If $\frac{1}{n} \sum_{i \in S} \langle X_i, u \rangle^4 \leqslant O(1)$ for all $u \in \mathcal{E}_k(1)$, then $\frac{1}{n} \sum_{i \in S} \langle X_i^{\otimes 4}, U \rangle \leqslant O(1)$ for all $U \in \mathcal{P}_k$. Since we can efficiently optimize over $\mathcal{P}_k$, we get an efficiently computable estimator with error $O(\varepsilon^{3/4})$ for Gaussian distributions. Furthermore, if we first use a proper thresholding (that we discuss below), our sum-of-squares proof also works for heavy-tailed distributions, that, apart from the certifiably bounded 4-th moment (that we cannot avoid with the sum-of-squares approach), are only required to have entrywise bounded 8-th moment.

Robust sparse regression with heavy-tailed designs is much more challenging. Again, for simplicity assume that $0.1 \cdot \operatorname{Id} \preceq \Sigma \preceq 10 \cdot \operatorname{Id}$ and $\sigma = 1$. First, there is an issue even without the adversarial noise: $\|\nabla H_{[n]}^*\|_\infty$ can be very large. Even under bounded fourth moment assumption, it can have magnitude $\tilde{O}(d^{1/4}/n)$, which is too large in the sparse setting. Hence we have to perform an additional thresholding step and remove large entries of $X$. Usually thresholding of the design matrix should be

---

[8]These relaxations were also used in [dKNS20] in the context of sparse PCA, but they used them in a different way.

done very carefully since it breaks the relation between $X$ and $y$. [Sas22] required the thresholding parameter $\tau$ to be large enough and depend polynomially on $\|\beta^*\|$ so that this dependence does not break significantly. Since $\|\nabla H^*_{[n]}\|_\infty$ can be as large as $\tilde{O}(\tau/n)$, the sample complexity of their estimator also depends polynomially on $\|\beta^*\|$.

Our idea of thresholding is very different, and it plays a significant role in our analysis, especially in the proof of strong convexity. Since we already have to work with outliers chosen by the adaptive adversary, we know that for an $\varepsilon$-fraction of samples, the dependence of $y$ on $X$ can already be broken. So, if we choose the thresholding parameter $\tau$ to be large enough so that with high probability it only affects an $\varepsilon$-fraction of samples, we can simply treat the samples affected by such thresholding as additional adversarial outliers, and assume that the adaptive adversary corrupted $2\varepsilon n$ samples. Note that since $\mathcal{D}$ is heavy-tailed, each sample $X_i^*$ might have entries of magnitude $d^{\Omega(1)}$. However, $y$ depends only on the inner products $\langle X_i^*, \beta^* \rangle$, and this inner product depends only on the entries of $X^*$ that correspond to the support of $\beta^*$. Even though we don't know the support, we can guarantee that for $\tau \geqslant 20\sqrt{k/\varepsilon}$, all entries of $X_i$ from the support of $\beta^*$ are bounded by $\tau$ with probability $1 - \varepsilon/2$. Indeed, since the variance of each entry is bounded by 10, Chebyshev's inequality implies that this entry is smaller than $\tau$ with probability at least $1 - \varepsilon/(2k)$, and by union bound, $\langle X_i^*, \beta^* \rangle$ is not affected by the thresholding with probability $1 - \varepsilon/2$. Hence by Chernoff bound, with overwhelming probability, the number of samples affected by our thresholding is at most $\varepsilon n$.

Let us denote the distribution of the rows of $X^*$ after thresholding with parameter $\tau$ by $\mathcal{D}(\tau)$. After the thresholding step, we can assume that $X_i^* \stackrel{\text{iid}}{\sim} \mathcal{D}(\tau)$. Note that thresholding can shift the mean, i.e. $\mathbb{E}\, X_i^*$ can be nonzero. It is easy to see that $\|\mathbb{E}_{x \sim \mathcal{D}(\tau)}\, x\|_\infty \leqslant O(1/\tau)$. Hence by Bernstein's inequality, $\|\nabla H^*_{[n]}\|_\infty \leqslant \tilde{O}\left(\sqrt{1/n} + \tau/n + 1/\tau\right)$ with high probability[9]. In particular, in order to get the error bounded by $O(\varepsilon^{3/4})$, we need to take $\tau \gtrsim \sqrt{k}/\varepsilon^{3/4}$, and it affects sample complexity. Furthermore, our sum-of-squares proof requires that $\left\|\frac{1}{n}\sum_{i=1}^n (X_i^*)^{\otimes 4} - \mathbb{E}(X_1^*)^{\otimes 4}\right\|_\infty$ is smaller that $1/k^2$. It can be shown that this quantity is bounded by $\tilde{O}\left(\sqrt{1/n} + \tau^4/n + 1/\tau^4\right)$ with high probability[10]. In particular, we need $n \geqslant \tilde{O}(\tau^4 k^2)$, so for $\tau \gtrsim \sqrt{k}/\varepsilon^{3/4}$, we have to take $n \geqslant \tilde{O}(k^4/\varepsilon^3)$. As was discussed in Remark 1.9, if $\mathcal{D}$ has sub-exponential tails, we do not have to do the thresholding, and the bounds from [DKK+22] allow to avoid this dependence of $n$ on $\varepsilon$. Note that due to the SQ lower bound (Proposition 1.11), sample complexity $k^4$ is likely to be necessary, even for Gaussian designs.

Finally, let us discuss the strong convexity property. Here, we do not assume any properties related to sum-of-squares, and focus on the weak assumptions of Theorem 1.4. First, assume that we need to show strong convexity only for sparse vectors, and not for all $u \in \mathcal{E}_k(r)$. As was observed in prior works on regression with oblivious outliers, e.g. [dLN+21], $\rho(u) := H_{\hat{S}}(\beta^* + u) - H_{\hat{S}}(\beta^*) - \langle \nabla H_{\hat{S}}, u \rangle$ can be lower bounded by $\frac{1}{2}\sum_{i \in \hat{S}} \langle X_i, u \rangle^2 \mathbf{1}_{[|\langle X_i, u\rangle - y_i| \leqslant 1]} \mathbf{1}_{[|\langle X_i, u\rangle| \leqslant 1]}$. Let $C(u) = S_{\text{good}} \cap \hat{S} \cap A \cap B(u)$, where $A$ is the set of samples where $|\eta_i| \leqslant 1$ and $B(u) = \{i \in [n] \mid |\langle X_i, u\rangle| \leqslant 1\}$. Then, $\rho(u) \geqslant \Omega(\sum_{i \in C(u)} \langle X_i^*, u\rangle^2)$. It can be shown that for some suitable $r$ and for each $k$-sparse $u$ of norm $r$, $C(u)$ is a large subset of the set $A$ (of size at least $0.99|A|$). Note that since $A$ is *independent* of $X^*$, the rows of $X^*$ that correspond to indices from $A$ are just iid samples from $\mathcal{D}$. If $X_i^*$ were Gaussian, we could have applied concentration bounds and prove strong convexity via union bound argument over subsets of size $0.99|A|$. In the heavy-tailed case, we need a different argument. For a fixed set $C$ of size $0.99|A|$, we can use Bernstein's inequality[11]. We cannot use union bound argument over all subsets of size $0.99|A|$ (there are too many), but fortunately we do not need it since for each $k$-sparse $u$ of norm $r$, it is enough to show that $\sum_{i \in T(u)} \langle X_i^*, u\rangle^2 \geqslant \Omega(r^2)$, where $T(u) \subset A$ is the set of the smallest (in absolute value) $0.99|A|$ entries of the vector $X_A^* u \in \mathbb{R}^{|A|}$. Hence, we can use an epsilon-net argument for the set of $k$-sparse vectors $u$ (of norm $r$). This set has very dense nets of

---

[9]Here we used the fact that $\phi(\eta_i) \leqslant 2$.

[10][DKLP22] used thresholding for robust sparse mean estimation, and showed a similar bound for second-order tensors. We generalize it to higher order tensors.

[11]Using a standard truncation argument. See also Proposition C.1 of [PJL20] for a similar argument in the dense setting.

(relatively) small size, and this is enough to show the lower bound $\sum_{i \in C(u)} \langle X_i^*, u \rangle^2 \geqslant \Omega(r^2)$ for all $k$-sparse $u$ of norm $r$ with high probability, as long as $n \geqslant \tilde{O}(k^2)$.

In order to show the same bound for all $u \in \mathcal{E}_k(r)$ of norm $r$, we observe that[12] if a quadratic form is $\Theta(r^2)$ on $K$-sparse vectors of norm $r$ for some $K \gtrsim k$, then it is also $\Theta(r^2)$ on all $u \in \mathcal{E}_k(r)$, and applying the argument from the previous paragraph to $K$-sparse vectors, we get the desired bound. We remark that directly proving it for $u \in \mathcal{E}_k(r)$ is challenging, since we extensively used the properties of the set of sparse vectors that are not satisfied by $\mathcal{E}_k(r)$, e.g. the existence of very dense epsilon-nets of small size.

## 3 Future Work

There is an interesting open problem in robust sparse regression that is not captured by our techniques. For sparse mean estimation, in the Gaussian case, there exists a polynomial time algorithm with nearly optimal guarantees: It achieves error $O(\tilde{\varepsilon})$ with $k^4 \, \text{polylog}(d)/\varepsilon^2$ samples ([DKK+22]). This algorithm uses a sophisticated sum-of-squares program[13]. It is reasonable to apply the techniques of [DKK+22] to robust sparse regression in order to achieve nearly optimal error $O(\tilde{\varepsilon})$ with $\text{poly}(k)$ samples. However, simple approaches (e.g. our approach with replacing the sparse constraints by the elastic constraints) fail in this case. Here we provide a high-level explanation of the issue. In order to combine the filtering algorithm with their techniques, we need to check whether the values of a certain quartic form are small on all sparse vectors. The analysis in [DKK+22] shows that this form is indeed small for the uncorrupted sample with high probability (see their Lemma E.2.). Since we want the filtering algorithm to be efficient, we have to use a *relaxation* of sparse vectors. Hence we need to find a sum-of-squares (or some other nice relaxation) version of the proof from [DKK+22]. However, in their proof they use a *covering argument*, and it is not clear how to avoid it. This argument fails for reasonable relaxations that we have thought about. Both potential outcomes (either an algorithm or a computational lower bound) are interesting: An algorithm would likely require new sophisticated ideas, and a lower bound would show a significant difference between robust sparse regression and robust mean estimation, while, so far, the complexity pictures of these problems have seemed to be quite similar.

Another interesting direction is to get error $o(\sqrt{\varepsilon})$ for distributions that do not necessarily have certifiably bounded moments. As was shown in [HL19], only moment assumptions (without certifiability) are not enough for efficient robust mean estimation, and the same should be true also for linear regression. However, other assumptions on distribution $\mathcal{D}$ can make the problem solvable in polynomial time. For robust mean estimation, some symmetry assumptions are enough even for heavy-tailed distributions without the second moment[14] (see [NST23]). It is interesting to investigate what assumptions on the design distribution are sufficient for existence of efficiently computable estimators for robust sparse regression.

## Acknowledgments and Disclosure of Funding

Chih-Hung Liu is supported by Ministry of Education, Taiwan under Yushan Fellow Program with the grant number MOE-111-YSFEE-0003-006-P1 and by National Science and Technology Council, Taiwan with the grant number 111-2222-E-002-017-MY2.

---

[12]Similar arguments are sometimes used to prove the restricted eigenvalue property of random matrices.

[13]A similar program for the dense setting was studied in [KMZ22].

[14]And, in some sense, even without the first moment, if instead of the mean we estimate the center of symmetry.

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

# A Properties of the Huber loss minimizer

**Definition A.1.** For $w \in \mathbb{R}^n_{\geqslant 0}$, the *weighted Huber loss function* is defined as

$$H_w(\beta) = \sum_{i \in [n]} w_i h(\langle X_i, \beta \rangle - y_i) \text{ where } h(x_i) = \begin{cases} \frac{1}{2} x_i^2 & \text{if } |x_i| \leqslant 2; \\ 2|x_i| - 2 & \text{otherwise.} \end{cases}$$

For a penalty parameter $\lambda$, the $\ell_1$-penalized Huber loss restricted to $S$ is defined as $L_w(\beta) := H_w(\beta) + \lambda \cdot \|\beta\|_1$.

**Lemma A.2.** *Suppose that $w \in \mathbb{R}^n_{\geqslant 0}$, $u \in \mathbb{R}^d$, $\gamma_1, \gamma_2, \lambda > 0$ satisfy the following properties:*

1. $\left| \sum_{i=1}^n w_i \phi(\eta_i + \zeta_i) \langle X_i''(\tau), u \rangle \right| \leqslant \gamma_1 \|u\|_1 + \gamma_2 \|\Sigma^{1/2} u\|$,

2. $\lambda \geqslant 2\gamma_1$,

3. $H_w(\beta^* + u) + \lambda \cdot \|\beta^* + u\|_1 \leqslant H_w(\beta^*) + \lambda \cdot \|\beta^*\|_1$.

*Then*

$$\|u\|_1 \leqslant \left( 4\sqrt{k/\sigma_{\min}} + 2\gamma_2/\lambda \right) \cdot \left\| \Sigma^{1/2} u \right\|,$$

*where $\sigma_{\min}$ is the minimal eigenvalue of $\Sigma$.*

*Proof.* Let $\mathcal{K} = \operatorname{supp}(\beta^*)$. Note that

$$\|\beta^* + u\|_1 = \left\| \beta^* + u_{\overline{\mathcal{K}}} + u_{\mathcal{K}} \right\|_1 \geqslant \|\beta^*\|_1 + \left\| u_{\overline{\mathcal{K}}} \right\|_1 - \|u_{\mathcal{K}}\|_1.$$

By the convexity of $H_w$,

$$H_w(\beta^* + u) - H_w(\beta^*) \geqslant -\left| \sum_{i=1}^n w_i \phi(\eta_i + \zeta_i) \langle X_i''(\tau), u \rangle \right| \geqslant -\lambda \|u\|_1/2 - \gamma_2 \left\| \Sigma^{1/2} u \right\|.$$

Hence

$$\begin{aligned}
0 &\geqslant \lambda \cdot \left( \|\beta^* + u\|_1 - \|\beta^*\|_1 \right) + H_w(\eta + \zeta + Xu) - H_w(\eta + \zeta) \\
&\geqslant \lambda \cdot \left( \left\| u_{\overline{\mathcal{K}}} \right\|_1 - \|u_{\mathcal{K}}\|_1 \right) - \tfrac{1}{2} \lambda \cdot \|u_{\mathcal{K}}\|_1 - \tfrac{1}{2} \lambda \cdot \left\| u_{\overline{\mathcal{K}}} \right\|_1 - \gamma_2 \\
&\geqslant \tfrac{1}{2} \lambda \cdot \left\| u_{\overline{\mathcal{K}}} \right\|_1 - \tfrac{3}{2} \lambda \|u_{\mathcal{K}}\|_1 - \gamma_2.
\end{aligned}$$

Therefore,

$$\lambda \|u\|_1 \leqslant 4\lambda \|u_{\mathcal{K}}\|_1 + 2\gamma_2 \left\| \Sigma^{1/2} u \right\| \leqslant 4\lambda \sqrt{k} \|u\| + 2\gamma_2 \left\| \Sigma^{1/2} u \right\| \leqslant 4\lambda \sqrt{\frac{k}{\sigma_{\min}}} \left\| \Sigma^{1/2} u \right\| + 2\gamma_2 \left\| \Sigma^{1/2} u \right\|.$$

$\square$

**Theorem A.3.** *Let $\rho, \gamma_1, \gamma_2 > 0$ and*

$$r = 100 \cdot \left( \frac{\lambda \sqrt{k/\sigma_{\min}}}{\rho} + \frac{\gamma_2}{\rho} \right),$$

*where $\sigma_{\min}$ is the minimal eigenvalue of $\Sigma$. Let $k' \geqslant 100k/\sigma_{\min}$. Consider the $k'$-elastic ellipsoid of radius $r$:*

$$\mathcal{E}_{k'}(r) = \left\{ u \in \mathbb{R}^d \mid \|\Sigma^{1/2} u\| \leqslant r, \|u\|_1 \leqslant \sqrt{k'} \cdot r \right\}.$$

*Suppose that the weights $w \in \mathbb{R}^n$ are such that the following two properties hold:*

1. *Gradient bound: For all $u \in \mathcal{E}_{k'}(r)$,*

$$\left| \sum_{i=1}^n w_i \phi(\eta_i + \zeta_i) \langle X_i''(\tau), u \rangle \right| \leqslant \gamma_1 \|u\|_1 + \gamma_2 \left\| \Sigma^{1/2} u \right\|,,$$

2. *Strong convexity on the boundary: For all $u \in \mathcal{E}_{k'}(r)$ such that $\left\|\Sigma^{1/2}u\right\| = r$,*

$$H_w(\beta^* + u) - H_w(\beta^*) \geqslant -\left|\sum_{i=1}^{n} w_i \phi(\eta_i + \zeta_i)\langle X_i''(\tau), u\rangle\right| + \rho \cdot r^2 \,.$$

*Let*

$$\lambda \geqslant 2\gamma_1 + \gamma_2 \cdot \sqrt{\frac{\sigma_{\min}}{k}} \,.$$

*Then the minimizer $\hat{\beta}$ of the weighted penalized Huber loss with penalty $\lambda$ and weights $w$ satisfies*

$$\left\|\Sigma^{1/2}\left(\hat{\beta} - \beta^*\right)\right\| < r \,.$$

*Proof.* Let $\hat{u} = \hat{\beta} - \beta^*$. If $\left\|\Sigma^{1/2}\hat{u}\right\| < r$, we get the desired bound. Otherwise, let $u$ be the (unique) point in the intersection of $\partial\mathcal{E}_{k'}(r)$ and the segment $[0, \hat{u}] \subset \mathbb{R}^d$. By convexity of the penalized loss,

$$H_w(\eta + \zeta + Xu) + \lambda \cdot \|\beta^* + u\|_1 \leqslant H_w(\eta + \zeta) + \lambda \cdot \|\beta^*\|_1 \,,$$

Since $u \in \partial\mathcal{E}_{k'}(r)$, either $\left\|\Sigma^{1/2}u\right\| = r$, or $\|u\|_1 = \sqrt{k'} \cdot r$. Let us show that the latter is not possible. Since $\lambda \geqslant 2\gamma_1$, we can apply Lemma A.2:

$$\sqrt{k'} \cdot r = \left(4\sqrt{k/\sigma_{\min}} + 2\gamma_2/\lambda\right) \cdot r \,.$$

Cancelling $r$ and using the bound $\lambda \geqslant \gamma_2 \cdot \sqrt{\frac{\sigma_{\min}}{k}}$, we get a contradiction. Hence $\left\|\Sigma^{1/2}u\right\| = r$. By the strong convexity and the gradient bound,

$$\begin{aligned} H_w(\beta^* + u) - H_w(\beta^*) &\geqslant -\left|\sum_{i=1}^{n} w_i \phi(\eta_i + \zeta_i)\langle X_i''(\tau), u\rangle\right| + \rho \cdot \left\|\Sigma^{1/2}u\right\|^2 \\ &\geqslant \rho \cdot r^2 - \tfrac{1}{2}\lambda \cdot \|u\|_1 - \gamma_2\left\|\Sigma^{1/2}u\right\| \\ &= \rho \cdot r^2 - \tfrac{1}{2}\lambda \cdot \|u\|_1 - \gamma_2 r \,. \end{aligned}$$

Note that
$$H_w(\beta^* + u) - H_w(\beta^*) \leqslant \lambda \cdot \left(\|\beta^*\|_1 - \|\beta^* + u\|_1\right) \leqslant \lambda\|u\|_1 \,.$$
By putting the above two inequality together and by Lemma A.2 , we have that

$$\rho \cdot r^2 \leqslant \tfrac{3}{2}\lambda\|u\|_1 + \gamma_2 r \leqslant 6\lambda\sqrt{k/\sigma_{\min}} \cdot r + 5\gamma_2 r \,.$$

Dividing both sides by $\rho \cdot r$, we get

$$r < 100 \cdot \left(\frac{\lambda\sqrt{k/\sigma_{\min}}}{\rho} + \frac{\gamma_2}{\rho}\right) \,,$$

a contradiction. Therefore, $\left\|\Sigma^{1/2}\hat{u}\right\| < r$.

$\square$

## B  Heavy-tailed Designs

First, we define a bit more general model than Definition 1.1

**Definition B.1** (Robust Sparse Regression with 2 Adversaries)**.** Let $n, d, k \in \mathbb{N}$ such that $k \leqslant d$, $\sigma > 0$, $\alpha \in (0, 1]$ and $\varepsilon \lesssim \alpha$. Let $\mathcal{D}$ be a probability distribution in $\mathbb{R}^d$ with mean 0 and covariance

$\Sigma$. Let $y^* = X^*\beta^* + \eta$, where $X$ is an $n \times d$ random matrix with rows $X_i^* \overset{\text{iid}}{\sim} \mathcal{D}$, $\beta^* \in \mathbb{R}^d$ is $k$-sparse, $\eta \in \mathbb{R}^n$ is independent of $X^*$ and has at least $\alpha \cdot n$ entries bounded by $\sigma$ in absolute value[15].

An instance of our model is a pair $(X, y)$, where $X \in \mathbb{R}^{n \times d}$ is a matrix and $y \in \mathbb{R}^n$ is a vector such that there exists a set $S_{\text{good}} \subseteq [n]$ of size at least $(1 - \varepsilon)n$ such that for all $i \in S_{\text{good}}$, $X_i = X_i^*$ and $y_i = y_i^*$.

**Definition B.2.** Let $M > 0$, $t \in \mathbb{N}$, and let $\ell \geqslant 2t$ be an even number. We say that a probability distribution $\mathcal{D}$ in $\mathbb{R}^d$ with zero mean and covariance $\Sigma$ has $\ell$-*certifiably $M$-bounded* $(2t)$-*th moment*, if there exist polynomials $h_1, \ldots, h_m \in \mathbb{R}[u_1, \ldots, u_d]$ of degree at most $\ell/2$ such that

$$\underset{x \sim \mathcal{D}}{\mathbb{E}} \langle x, u \rangle^{2t} + \sum_{i=1}^{m} h_i^2(u) = M^{2t} \cdot \|\Sigma\|^t \cdot \|u\|^{2t} .$$

In this section we prove the following theorem

**Theorem B.3** (Heavy-tailed designs, general formulation). *Let $n, d, k, X, y, \varepsilon, \mathcal{D}, \Sigma, \sigma, \alpha$ be as in Definition B.1, and let $\delta \in (0, 1)$.*

*Suppose that for some $s > 2$, $t \in \mathbb{N}$, $M_s, M_{2t} \geqslant 1$, and even number $\ell \geqslant 2t$, $\mathcal{D}$ has $M_s$-bounded $s$-th moment, and $\ell$-certifiably $M_{2t}$-bounded $(2t)$-th moment. In addition, $\mathcal{D}$ has entrywise $\nu$-bounded $(4t)$-th moment.*

*There exists an algorithm that, given $X, y, k, \varepsilon, \sigma, M_{2t}, \ell, t, \delta$ and $\hat{\sigma}_{\max}$ such that $\|\Sigma\| \leqslant \hat{\sigma}_{\max} \leqslant O(\|\Sigma\|)$, in time $(n + d)^{O(\ell)}$ outputs $X' \in \mathbb{R}^{n \times d}$ and weights $w = (w_1, \ldots, w_n)$ such that if*

$$n \gtrsim \frac{10^{10t} \left( M_{2t}^{2t} \cdot \nu^{4t} + \left( 10^5 M_s \right)^{\frac{2s}{s-2}} \right) \cdot \left( \kappa(\Sigma)^{4+s/(s-2)} + \kappa(\Sigma)^{2t} \right)}{\varepsilon^{2t-1}} \cdot k^{2t} \log(d/\delta)$$

*then with probability at least $1 - \delta$, the weighted $\ell_1$-penalized Huber loss estimator $\hat{\beta}_w = \hat{\beta}_w(X', y)$ with weights $w$ (as in Definition 2.1) and parameter $h$ satisfies*

$$\left\| \Sigma^{1/2} \left( \hat{\beta}_w - \beta^* \right) \right\| \leqslant O\left( \frac{M_{2t} \sqrt{\kappa(\Sigma)}}{\alpha} \cdot \sigma \cdot \varepsilon^{1 - \frac{1}{2t}} \right).$$

Let us explain how this result implies Theorem 1.4 and Theorem 1.7.

Theorem 1.4 is a special case of Theorem B.3 with $t = 1$, $s = 3$, $\ell = 2$, $M_{2t} = 1$, $M_s = M$, $\alpha = 0.01$. Indeed, we only need to estimate $\|\Sigma\|$ up to a constant factor. We can do it by estimating the variance of the first coordinate of $x \sim \mathcal{D}$. Applying median-of-means algorithm[16] to the first coordinate, we get an estimator $\tilde{\sigma}^2$ that is $O(\nu^2 \|\Sigma\| \sqrt{\varepsilon})$-close to the variance of the first coordinate $\sigma_1^2$. Note that $\|\Sigma\|/\kappa(\Sigma) \leqslant \sigma_1^2 \leqslant \|\Sigma\|$. Since in Theorem 1.4 $\kappa(\Sigma)$ and $\nu$ are constants, and $\varepsilon$ is sufficiently small, we get that $\frac{1}{2\kappa(\Sigma)} \|\Sigma\| \leqslant \tilde{\sigma}_{\max}^2 \leqslant 2\|\Sigma\|$. Hence for a constant $C \geqslant \kappa(\Sigma)$, $\hat{\sigma}_{\max} = 2C\tilde{\sigma}^2$ is the desired estimator of $\|\Sigma\|$.

Similarly, Theorem 1.7 is a special case of Theorem B.3 with $t = 2$, $s = 4$, $M_{2t} = M_s = M$, $\alpha = 0.01$. $\|\Sigma\|$ can be estimated using the procedure described above.

Before proving the theorem, note that we can without loss of generality assume that $\sigma = 1$. Indeed, since $\sigma$ is known, we can simply divide $X$ and $y$ by it before applying the algorithm.

## B.1 Truncation

We cannot work with $X^*$ directly since it might have very large values, and Bernstein inequality that we use for random vectors concentration would give very bad bounds if we work with $X^*$. Fortunately,

---

[15]Our result also works for more general model, where we require $\alpha n$ entries to be bounded by $\sigma$ for some $\alpha \gtrsim \varepsilon$. The error bound in this case also depends on $\alpha$.

[16]See, for example, Fact 2.1. from [DKLP22], where they state the guarantees of the median-of-means algorithm.

we can perform truncation. This technique was used in [DKLP22] for sparse mean estimation and in [Sas22] for sparse regression.

For $\tau > 0$ let $X'_{ij}(\tau) = X^*_{ij}\mathbf{1}_{\left[|X^*_{ij}|\leqslant\tau\right]}$. Note that since $\mathbb{P}\left[|X^*_{ij}| > \tau\right] \leqslant \|\Sigma\|/\tau^2$, if $\tau \gtrsim \sqrt{\|\Sigma\|k/\varepsilon}$, then the number of entries $i$ where $\left\langle X^*_i, \beta^*\right\rangle \neq \left\langle X'_i(\tau), \beta^*\right\rangle$ is at most $\varepsilon n$ with probability at least $1 - 2^{-\varepsilon n/10} \geqslant 1 - \delta/10$. Hence in the algorithm we assume that the input is $X'(\tau)$ instead of $X^*$, and we treat the entries where $\left\langle X^*_i, \beta^*\right\rangle \neq \left\langle X'_i(\tau), \beta^*\right\rangle$ as corrupted by an adversary.

Concretely, further we assume that we are given $\left\{\left(X''_i(\tau), y_i, w_i\right)\right\}_{i=1}^n$ such that $y = X'(\tau)\beta^* + \eta + \zeta$, where $X''(\tau) \in \mathbb{R}^{n\times d}$ differs from $X'(\tau) \in \mathbb{R}^{n\times d}$ only in rows from the set $S_{\text{bad}} \subset [n]$ of size at most $\tilde{\varepsilon} n$ (where $\varepsilon \leqslant \tilde{\varepsilon} \leqslant O(\varepsilon)$), $\zeta \in \mathbb{R}^n$ is an $\tilde{\varepsilon} n$-sparse vector such that $\text{supp}(\zeta) \subseteq S_{\text{bad}}$, $\beta^* \in \mathbb{R}^d$ a $k$-sparse vector, and $\eta \in \mathbb{R}^n$ is oblivious noise such that at least $\alpha n$ entries do not exceed 1 in absolute value.

In addition, we define

$$\mathcal{W}_{\tilde{\varepsilon}} = \left\{w \in \mathbb{R}^n \;\middle|\; \forall i \in [n] \;\; 0 \leqslant w_i \leqslant 1/n, \;\; \sum_{i=1}^n w_i \geqslant (1 - \tilde{\varepsilon})n\right\}.$$

The weights for the Huber loss will be from $\mathcal{W}_{\tilde{\varepsilon}}$.

Appendix E will discuss more properties of the truncation.

## B.2 Gradient Bound

**Lemma B.4.** *Let $b, \gamma_1 > 0$. Suppose that $w \in \mathcal{W}_{\tilde{\varepsilon}}$ and $u \in \mathbb{R}^d$ satisfy*

$$\sum_{i\in[n]} w_i\left\langle X''_i(\tau), u\right\rangle^{2t} \leqslant b^{2t} \cdot \left\|\Sigma^{1/2}u\right\|^{2t},$$

$$\frac{1}{n}\sum_{i\in[n]} \left\langle X'_i(\tau), u\right\rangle^{2t} \leqslant b^{2t} \cdot \left\|\Sigma^{1/2}u\right\|^{2t},$$

*and*

$$\left\|\frac{1}{n}\sum_{i\in[n]} \phi(\eta_i)X'_i(\tau)\right\|_\infty \leqslant \gamma_1 .$$

*Then*

$$\left|\sum_{i=1}^n w_i\phi(\eta_i + \zeta_i)\left\langle X''_i(\tau), u\right\rangle\right| \leqslant \gamma_1 \cdot \|u\|_1 + 6 \cdot b\left\|\Sigma^{1/2}u\right\|^{2t} \cdot \tilde{\varepsilon}^{1-\frac{1}{2t}} .$$

*Proof.* Denote $F(w) = \sum_{i\in[n]}(1/n - w_i)\phi(\eta_i)\left\langle X'_i(\tau), u\right\rangle$. It is a linear function of $w$, so $|F(w)|$ is maximized in one of the vertices of the polytope $\mathcal{W}_{\tilde{\varepsilon}}$. This vertex corresponds to set $S_w$ of size at least $(1 - \tilde{\varepsilon})n$. That is, the weights of the entries from $S_w$ are $1/n$, and outside of $S_w$ the weights are zero. It follows that

$$\left|\sum_{i=1}^n w_i\phi(\eta_i + \zeta_i)\left\langle X''_i(\tau), u\right\rangle\right|$$

$$\leqslant \left|\sum_{i\in[n]} w_i\phi(\eta_i)\left\langle X'_i(\tau), u\right\rangle\right| + \left|\sum_{i\in S_{\text{bad}}} w_i\phi(\eta_i)\left\langle X'_i(\tau), u\right\rangle\right| + \left|\sum_{i\in S_{\text{bad}}} w_i\phi(\eta_i + \zeta_i)\left\langle X''_i(\tau), u\right\rangle\right| \quad \text{(Triangle Inequality)}$$

$$\leqslant \gamma_1 \cdot \|u\|_1 + 2\sum_{i\in S_w} \frac{1}{n}\left|\left\langle X'_i(\tau), u\right\rangle\right| + 2\sum_{i\in S_{\text{bad}}} \frac{1}{n}\left|\left\langle X'_i(\tau), u\right\rangle\right| + 2\sum_{i\in S_{\text{bad}}} w_i\left|\left\langle X''_i(\tau), u\right\rangle\right|$$

$$\leqslant \gamma_1 \cdot \|u\|_1 + 4 \cdot \tilde{\varepsilon}^{1-\frac{1}{2t}} \cdot \left( \sum_{i\in[n]} \tfrac{1}{n} \langle X_i'(\tau), u \rangle^{2t} \right)^{\frac{1}{2t}} + 2\tilde{\varepsilon}^{1-\frac{1}{2t}} \cdot \left( \sum_{i\in[n]} w_i \langle X_i''(\tau), u \rangle^{2t} \right)^{\frac{1}{2t}} \qquad \text{(Hölder's inequality)}$$

$$\leqslant \gamma_1 \cdot \|u\|_1 + 6 \cdot \tilde{\varepsilon}^{1-\frac{1}{2t}} \cdot b \left\| \Sigma^{1/2} u \right\|^{2t} .$$

□

The following lemma provides a bound on $\gamma_1$:

**Lemma B.5.** *With probability at least* $1 - \delta/10$,

$$\left\| \tfrac{1}{n} \sum_{i\in[n]} \phi(\eta_i) X_i'(\tau) \right\|_\infty \leqslant 10\sqrt{\|\Sigma\| n \log(d/\delta)} + 10\tau \cdot \log(d/\delta) + 2n \cdot \|\Sigma\|/\tau .$$

*Proof.* It follows from Bernstein's inequality Fact I.1 and the fact that

$$\tfrac{1}{n} \sum_{i\in[n]} |\phi(\eta_i)| \cdot \left| \mathbb{E}\, X_i'(\tau) \right| \leqslant 2\,\mathbb{E}\, X_1'(\tau) \leqslant 2n \cdot \|\Sigma\|/\tau ,$$

where we used Corollary E.4. □

### B.2.1 Strong Convexity

**Lemma B.6.** *Suppose that* $\alpha \geqslant 1000\tilde{\varepsilon}$, $\tau \gtrsim 1000 \cdot v^2 \cdot \|\Sigma\| \sqrt{k''} / (r\sqrt{\sigma_{\min}})$ *and*

$$n \gtrsim \left( (k'')^2 \log d + k'' \log(1/\delta) \right) 10^{5s/(s-2)} M_s^{s/(s-2)} \kappa(\Sigma)^{2+s/(s-2)} / \alpha ,$$

*where* $k'' = 10^4 \cdot k' \cdot \sqrt{\|\Sigma\|}$. *Then with probability* $1 - \delta/10$, *for all* $u \in \mathcal{E}_{k'}(r)$ *such that* $\left\| \Sigma^{1/2} u \right\| = r$,

$$H(\beta^* + u) - H(\beta^*) \geqslant - \left| \sum_{i=1}^n w_i \phi(\eta_i + \zeta_i) \langle X_i, u \rangle \right| + \tfrac{1}{4} \cdot r^2 .$$

*Proof.* Denote $A_{\text{good}} = S_{\text{good}} \cap \mathcal{A}$, where $\mathcal{A}$ is a set of entries $i$ such that $|\eta_i| \leqslant 1$. Note that $\mathcal{A}$ is independent of $X^*$. It follows that

$$H(\beta^* + u) - H(\beta^*) - \sum_{i=1}^n w_i \phi(\eta_i + \zeta_i) \langle X_i, u \rangle \geqslant \tfrac{1}{2} \sum_{i=1}^n w_i \langle X_i, u \rangle^2 \mathbf{1}_{[|\eta_i + \zeta_i| \leqslant 1]} \mathbf{1}_{[|\langle X_i, u \rangle| \leqslant 1]}$$

$$\geqslant \tfrac{1}{2} \sum_{i\in S_{\text{good}}} w_i \langle X_i'(\tau), u \rangle^2 \mathbf{1}_{[|\eta_i| \leqslant 1]} \mathbf{1}_{[|\langle X_i'(\tau), u \rangle| \leqslant 1]}$$

$$\geqslant \tfrac{1}{2} \sum_{i\in A_{\text{good}}} w_i \langle X_i'(\tau), u \rangle^2 \mathbf{1}_{[|\langle X_i'(\tau), u \rangle| \leqslant 1]}$$

$$\geqslant \tfrac{1}{2} \sum_{i\in A_{\text{good}}} w_i \langle \tilde{X}_i, u \rangle^2 \mathbf{1}_{[|\langle \tilde{X}_i, u \rangle| \leqslant 1]} ,$$

where $\tilde{X}_i = \mathbf{1}_{\left[ \|X_i'(\tau)\| \leqslant 10^{5s/(s-2)} \cdot M_s^{s/(s-2)} \sqrt{\|\Sigma\| \cdot k''} \right]} X_i'(\tau)$.

Denote $F(w) = \sum_{i\in A_{\text{good}}} w_i \langle \tilde{X}_i, u \rangle^2 \mathbf{1}_{[|\langle \tilde{X}_i, u \rangle| \leqslant 1]}$. It is a linear function of $w$, so it is maximized in one of the vertices of the polytope $\mathcal{W}_{\tilde{\varepsilon}}$. This vertex corresponds to set $S_w$ of size at least $(1 - \tilde{\varepsilon})n$. That is, the weights of the entries from $S_w$ are $1/n$, and outside of $S_w$ the weights are zero.

$$\sum_{i\in A_{\text{good}}} w_i \langle \tilde{X}_i, u \rangle^2 \mathbf{1}_{[|\langle \tilde{X}_i, u \rangle| \leqslant 1]} \geqslant \tfrac{1}{n} \sum_{i\in A_{\text{good}} \cap S_w} \langle \tilde{X}_i, u \rangle^2 \mathbf{1}_{[|\langle \tilde{X}_i, u \rangle| \leqslant 1]} .$$

Hence we need a lower bound for $\sum_{i\in A(u)}\langle \tilde{X}_i, u\rangle^2$, where

$$A(u) = A_{\text{good}} \cap S_w \cap \left\{ i \in [n] \,\middle|\, |\langle \tilde{X}_i, u\rangle| \leq 1 \right\}.$$

In order to bound $\sum_{i\in A(u)}\langle \tilde{X}_i, u\rangle^2$ for vectors $u$ from the elastic ball $\mathcal{E}_{k'}(r)$, we first show that it is bounded for $k''$-sparse vectors $u'$ for some large enough $k''$. First we need to show that $\sum_{i\in\mathcal{I}}\langle \tilde{X}_i, u'\rangle^2$ is well-concentrated for a fixed set $\mathcal{I}$. Concretely, we need the following lemma:

**Lemma B.7.** *Suppose that $\tau \geq 1000 \cdot M_{2t} \cdot v^2 \cdot \|\Sigma\|\sqrt{k''}/(r\sqrt{\sigma_{\min}})$ for some $k'' \in \mathbb{N}$. Then for a fixed (independent of $\tilde{X}$) set $\mathcal{I}$ of size*

$$|\mathcal{I}| \gtrsim \left((k'')^2 \log d + k'' \log(1/\delta)\right) 10^{10s/(s-2)} M_s^{2s/(s-2)} \kappa(\Sigma)^{2+2s/(s-2)}$$

*and for all $k''$-sparse vectors $u' \in \mathbb{R}^d$ such that $r \leq \|\Sigma^{1/2}u'\| \leq 2r$,*

$$0.99 \cdot \|\Sigma^{1/2}u'\|^2 \leq \tfrac{1}{|\mathcal{I}|} \sum_{i\in\mathcal{I}}\langle \tilde{X}_i, u'\rangle^2 \leq 1.01 \cdot \|\Sigma^{1/2}u'\|^2 .$$

*with probability at least $1 - \delta$.*

*Proof.* First let us show that

$$0.995 \cdot \mathbb{E}\langle X_i^*, u'\rangle^2 \leq \mathbb{E}\langle \tilde{X}_i, u'\rangle^2 \leq 1.005 \cdot \mathbb{E}\langle X_i^*, u'\rangle^2 .$$

Since for each set $\mathcal{K}$ of size $k''$, $\mathbb{E}\left\|\left(X_i'(\tau)\right)_{\mathcal{K}}\right\|^2 = \sum_{j\in\mathcal{K}}\mathbb{E}\left(X_{ij}'(\tau)\right)^2 \leq 2\|\Sigma\|k''$ , by Markov's inequality,

$$\mathbb{P}\left[\left\|\left(X_i'(\tau)\right)_{\mathcal{K}}\right\|^2 > 10^{10s/(s-2)} \cdot M_s^{2s/(s-2)} \cdot \kappa(\Sigma)^{s/(s-2)}\|\Sigma\| \cdot k''\right] \leq \frac{1}{10^{10s/(s-2)} \cdot M_s^{2s/(s-2)} \cdot \kappa(\Sigma)^{s/(s-2)}} .$$

Denote $B = 10^{5s/(s-2)} \cdot M_s^{s/(s-2)} \cdot \kappa(\Sigma)^{s/(2s-4)}\sqrt{\|\Sigma\| \cdot k''}$. By Hölder's inequality, for all vectors $u' \in \mathbb{R}^d$ with support $\mathcal{K}$,

$$
\begin{aligned}
\mathbb{E}\langle X_i'(\tau), u'\rangle^2 &= \mathbb{E}\langle X_i(\tau), u'\rangle^2 \mathbf{1}_{[\|(X_i'(\tau))_{\mathcal{K}}\| \leq B]} + \mathbb{E}\langle X_i'(\tau), u'\rangle^2 \mathbf{1}_{[\|(X_i'(\tau))_{\mathcal{K}}\| > B]} \\
&\leq \mathbb{E}\langle \tilde{X}_i, u'\rangle^2 + 2\mathbb{E}\langle X_i^*, u'\rangle^2 \mathbf{1}_{[\|(X_i'(\tau))_{\mathcal{K}}\| > B]} + 2\mathbb{E}\langle X_i'(\tau) - X_i^*, u'\rangle^2 \\
&\leq \mathbb{E}\langle \tilde{X}_i, u'\rangle^2 + 2\left(\mathbb{E}\,\mathbf{1}_{[\|(X_i'(\tau))_{\mathcal{K}}\| > B]}\right)^{1-\frac{2}{s}} \cdot \left(\mathbb{E}\langle X_i^*, u'\rangle^s\right)^{\frac{2}{s}} + \frac{2v^4\|\Sigma\|^2 k''\|u'\|^2}{\tau^2} \\
&\leq \mathbb{E}\langle \tilde{X}_i, u'\rangle^2 + 2\frac{\|\Sigma\| \cdot \|u\|^2}{10^{10} \cdot \kappa(\Sigma)} + 2r^2/10^6 \\
&\leq \mathbb{E}\langle \tilde{X}_i, u'\rangle^2 + 2r^2/10^{10} + 2r^2/10^6 .
\end{aligned}
$$

where we used Lemma E.1 and the fact that $\|u'u'^\top\|_1 \leq k''\|u'u'^\top\| \leq k''\|u\|^2$. By Corollary E.5, $\mathbb{E}\langle X_i^*, u'\rangle^2 - 2r^2/10^6 \leq \cdot \mathbb{E}\langle X_i'(\tau), u'\rangle^2 \leq \mathbb{E}\langle X_i^*, u'\rangle^2 + 2r^2/10^6$. Hence

$$0.995\cdot\mathbb{E}\langle X_i^*, u'\rangle^2 \leq 0.999\cdot\mathbb{E}\langle X_i'(\tau), u'\rangle^2 \leq \mathbb{E}\langle \tilde{X}_i, u'\rangle^2 \leq 1.001\cdot\mathbb{E}\langle X_i'(\tau), u'\rangle^2 \leq 1.005\cdot\mathbb{E}\langle X_i^*, u'\rangle^2 .$$

For a fixed set $\mathcal{K}$ of size $k''$ and for all unit vectors $u' \in \mathbb{R}^d$ with support $\mathcal{K}$, by Bernstein inequality for covariance Fact I.2, with probability $1 - \delta$,

$$
\begin{aligned}
\left|\tfrac{1}{|\mathcal{I}|}\sum_{i\in\mathcal{I}}\langle \tilde{X}_i, u'\rangle^2 - \mathbb{E}\langle \tilde{X}_i, u'\rangle^2\right| &\leq 1000 \cdot \left(\sqrt{\frac{\|\Sigma\|B^2\log(d/\delta)}{|\mathcal{I}|}} + \frac{B^2\log(d/\delta)}{|\mathcal{I}|}\right) \cdot \|u'\|^2 \\
&\leq 4000 \cdot \left(\sqrt{\frac{\|\Sigma\|B^2\log(d/\delta)}{\sigma_{\min}^2|\mathcal{I}|}} + \frac{B^2\log(d/\delta)}{\sigma_{\min}|\mathcal{I}|}\right) \cdot r^2 .
\end{aligned}
$$

In order to make this quantity smaller than $r^2/1000$, it is sufficient to take $|\mathcal{I}| \gtrsim 10^{10s/(s-2)} M_s^{2s/(s-2)} \kappa(\Sigma)^{2+s/(s-2)} \cdot k'' \log(d/\delta)$.

By union bound over all subsets $\mathcal{K}$ of $[d]$ of size $k''$, we get the desired bound. $\qquad\square$

Let us bound the size of $A(u)$. $A_{\text{good}} \cap S_w$ has size at least $(\alpha - 3\tilde{\varepsilon})n \geqslant 0.997\alpha n$. By Lemma B.7 and Lemma F.2, $\left\|\tilde{X}u\right\|^2 \leqslant 1.1 \cdot \alpha n r^2$, hence at most $3\alpha n r^2/h \leqslant 0.001\alpha n$ entries of $\tilde{X}u$ can be greater than $h/2$. Therefore, $|A(u)| \geqslant 0.99\alpha n$.

Let $k'' = \min\left\{\lceil 10^4 k'\|\Sigma\|\rceil, d\right\}$. Recall that $\mathcal{A}$ a set of entries $i$ such that $|\eta_i| \leqslant 1$, and $\mathcal{A}$ is independent of $X^*$. By union bound, the result of Lemma B.7 also holds for all sets $\mathcal{I}$ that correspond to the bottom $0.99$-fraction of entries of vectors $(\tilde{X}u'')_{\mathcal{A}}$, where $u''$ are from an $(1/n^{10})$-net $\mathcal{N}$ in the set of all $k''$-sparse vectors $u'$ such that $\left\|\Sigma^{1/2}u'\right\| = 1.01r$. Let $u'$ be an arbitrary $k''$-sparse vector such that $\left\|\Sigma^{1/2}u'\right\| = 1.01r$, and let $u'' = u' + \Delta u$ be the closest vector in the net $\mathcal{N}$ to $u'$. It follows that

$$
\begin{aligned}
\sum_{i \in A(u)} \left\langle \tilde{X}_i, u' \right\rangle^2 &= \sum_{i \in A(u)} \left\langle \tilde{X}_i, u'' + \Delta u \right\rangle^2 \\
&\geqslant \sum_{i \in A(u)} \left\langle \tilde{X}_i, u'' \right\rangle^2 - 2n^3/n^{10} \\
&\geqslant 0.99\alpha n \cdot r^2 - 2n^{-7} \\
&\geqslant 0.9\alpha n \cdot r^2 \,.
\end{aligned}
$$

If $k'' = d$, we get the desired bound, since we can take $u' = u$. Otherwise, by Lemma B.7,

$$
\sum_{i \in A(u)} \left\langle \tilde{X}_i, u' \right\rangle^2 \leqslant \sum_{i \in \mathcal{A}} \left\langle \tilde{X}_i, u' \right\rangle^2 \leqslant 1.1 \cdot \alpha n \cdot r^2 \,,
$$

and we get the desired bound by Lemma F.2.

$\qquad\square$

## B.3 Putting everything together

First, we truncate the entries of $X$ and $X^*$ and obtain $X''(\tau)$ and $X'(\tau)$ using some $\tau$ such that

$$
\tau \gtrsim M_{2t}\sqrt{\|\Sigma\|} \cdot v^2 \cdot \sqrt{k''}/\varepsilon^{1-\frac{1}{2t}} \,,
$$

where $k'' = 10^6 \cdot k \cdot \kappa(\Sigma)$. We discuss the choice of $\tau$ further in this subsection. Let us denote $\tau' = \tau/\sqrt{\|\Sigma\|}$.

Then we find the weights $w_1, \ldots w_n$ using Algorithm C.1.

We will show all the conditions of Theorem A.3 are satisfied if

$$
n \geqslant C \cdot \frac{10^{10t}\left(M_{2t}^{2t} \cdot v^{4t} + \left(10^5 M_s\right)^{\frac{2s}{s-2}}\right) \cdot \left(\kappa(\Sigma)^{4+s/(s-2)} + \kappa(\Sigma)^{2t}\right)}{\varepsilon^{2t-1}} \cdot k^{2t} \log(d/\delta)
$$

for some large enough absolute constant $C$ and

$$
\lambda = 1000 \cdot M_{2t}\sqrt{\hat{\sigma}_{\max}} \cdot \varepsilon^{1-1/(2t)}/\sqrt{k} \geqslant 1000 \cdot \frac{M_{2t}\sqrt{\kappa(\Sigma)} \cdot \varepsilon^{1-1/(2t)}}{\sqrt{k/\sigma_{\min}}} \,.
$$

First let us show that the assumptions of Lemma B.4 are satisfied with $\gamma_1 \leqslant 100 \cdot M_{2t}\sqrt{\|\Sigma\|} \cdot \varepsilon^{1-1/(2t)}/\sqrt{k}$ and $\gamma_2 \leqslant 10M_{2t}\sqrt{\kappa(\Sigma)}$.

First we bound $\gamma_2$. Note that if $u \in \mathcal{E}_{k'}(r)$ for $k' = 100k/\sigma_{\min}$, then $\|u\|_1 \leqslant k''\|u\|$. Hence if

$$
n \geqslant 1000\left(v^{4t} \cdot (k'')^t + (\tau')^{2t}\right) \cdot (k'')^t \cdot t \log(d/\delta) \,,
$$

then Lemma D.2 implies that for all $u \in \mathcal{E}_{k'}(r)$, with probability $1 - \delta/10$,

$$\frac{1}{n} \sum_{i \in [n]} \langle X_i'(\tau), u \rangle^{2t} \leqslant \left( 2M_{2t} \sqrt{\|\Sigma\|} \right)^{2t} \cdot \|u\|^{2t} \leqslant \left( 2M_{2t} \sqrt{\kappa(\Sigma)} \right)^{2t} \cdot \left\| \Sigma^{1/2} u \right\|^{2t}.$$

Lemma D.2 and Lemma C.2 imply that for all $u \in \mathcal{E}_{k'}(r)$, with probability $1 - \delta/10$,

$$\sum_{i \in [n]} w_i \langle X_i''(\tau)(\tau), u \rangle^{2t} \leqslant \left( 2M_{2t} \sqrt{\kappa(\Sigma)} \right)^t \cdot \left\| \Sigma^{1/2} u \right\|^{2t}.$$

Let us bound $\gamma_1$. By Lemma B.5, if

$$n \geqslant 1000 \left( k \log(d/\delta)/\varepsilon^{2-1/t} + \tau \log(d/\delta) \sqrt{k'}/\varepsilon^{1-1/(2t)} \right),$$

then by with probability $1 - \delta/10$, $\gamma_1 \leqslant 100 \cdot \frac{M_{2t} \sqrt{\kappa(\Sigma)} \cdot \varepsilon^{1-1/(2t)}}{\sqrt{k/\sigma_{\min}}}$.

The strong convexity holds by Lemma B.6 with probability $1 - \delta/10$ as long as

$$n \gtrsim \left( k^2 \log(d/\delta) \right) 10^{5s/(s-2)} M_s^{s/(s-2)} \kappa(\Sigma)^{4+s/(s-2)}/\varepsilon,$$

where we used the fact that $\varepsilon \lesssim \alpha$ and that $\tau \gtrsim \sqrt{\|\Sigma\|} \cdot v^2 \cdot \sqrt{k''}/\varepsilon^{1-\frac{1}{2t}}$ satisfies the assumption of that lemma.

Therefore, all the conditions of Theorem A.3 are satisfied and we attain the desired bound of $O\left( \frac{M_{2t} \sqrt{\kappa(\Sigma)}}{\alpha} \cdot \varepsilon^{1-\frac{1}{2t}} \right)$ stated in Theorem B.3.

Now let us discuss the choice of $\tau$. First we can find an estimator $\hat{\kappa}$ of $\kappa(\Sigma)$ by plugging it into the formula

$$n = C \cdot \frac{10^{10t} \cdot \left( \hat{\kappa}^{4+s/(s-2)} + \hat{\kappa}^{2t} \right)}{\varepsilon^{2t-1}} \cdot k^{2t} \log(d/\delta).$$

Then we can take $\tau' = 0.01 \cdot \left( \frac{n}{\hat{\kappa}^t \cdot k^t \cdot t \log(d/\delta)} \right)^{1/(2t)}$. Note that if we express $n$ in terms of $\hat{\kappa}$ and plug into the formula for $\tau'$, we get that $\tau'$ is an increasing function of $\hat{\kappa}$. Also note that $\hat{\kappa} \geqslant \kappa(\Sigma)$. Hence both conditions are satisfied: $\tau := \sqrt{\hat{\sigma}_{\max}} \cdot \tau'$ is larger than the required lower bound for it, and $n$ is larger than $10000(\tau')^{2t} \cdot (k'')^t \log(d/\delta)$ and $10000\tau \log(d/\delta) \sqrt{k'}/\varepsilon^{1-1/(2t)}$ as required.

## C  Filtering

We use the following system of elastic constraints with sparsity parameter $K \geqslant 1$ and variables $v_1, \ldots, v_d, s_1, \ldots, s_d$:

$$\mathcal{A}_K : \begin{cases} \forall i \in [d] & s_i^2 = 1 \\ \forall i \in [d] & s_i v_i \geqslant v_i \\ \forall i \in [d] & s_i v_i \geqslant -v_i \\ & \sum_{i=1}^{d} v_i^2 \leqslant 1 \\ & \sum_{i=1}^{d} s_i v_i \leqslant \sqrt{K} \end{cases} \tag{C.1}$$

Note that the vectors from the elastic ball $\left\{ v \in \mathbb{R}^d \mid \|v\| \leqslant 1, \|v\|_1 \leqslant \sqrt{K} \right\}$ satisfy these constraints with $s_i = \text{sign}(v_i)$. We will later discuss the corresponding sum-of-squares certificates in Appendix D.

Let $a > 0$ be such that $\left\langle \frac{1}{n} \sum_{i=1}^{n} \left( X_i^* \right)^{\otimes 2t}, \tilde{\mathbb{E}} v^{\otimes 2t} \right\rangle \leqslant a^{2t}$.

> **Algorithm C.1** (Filtering algorithm).
>
> 1. Assign weights $w_1 = \ldots w_n = 1/n$.
>
> 2. Find a degree $2\ell$ pseudo-expectation $\tilde{\mathbb{E}}$ that satisfies $\mathcal{A}_K$ and maximizes $\left\langle \sum_{i=1}^{n} w_i X_i^{\otimes t}, \tilde{\mathbb{E}} v^{\otimes t} \right\rangle$.
>
> 3. If $\left\langle \frac{1}{n} \sum_{i=1}^{n} X_i^{\otimes 2t}, \tilde{\mathbb{E}} v^{\otimes 2t} \right\rangle < 10^t a^{2t}$, stop.
>
> 4. Compute $\tau_i = \left\langle X_i^{\otimes 2t}, \tilde{\mathbb{E}} v^{\otimes 2t} \right\rangle$ and reweight: $w_i' = (1 - \frac{\tau_i}{\|\tau\|_\infty}) \cdot w_i$.
>
> 5. goto 2.

**Lemma C.2.** *If at each step $\left\langle \frac{1}{n} \sum_{i=1}^{n} \left( X_i^* \right)^{\otimes 2t}, \tilde{\mathbb{E}} v^{\otimes 2t} \right\rangle \leqslant a^{2t}$, then the algorithm terminates in at most $\lceil 2\varepsilon n \rceil$ steps, and the resulting weights satisfy $\sum_{i=1}^{n} w_i \geqslant 1 - 2\varepsilon$.*

To prove it, we will use the following lemma:

**Lemma C.3.** *Assume that $\left\langle \frac{1}{n} \sum_{i=1}^{n} \left( X_i^* \right)^{\otimes 2t}, \tilde{\mathbb{E}} v^{\otimes 2t} \right\rangle \leqslant a^{2t}$, $\left\langle \frac{1}{n} \sum_{i=1}^{n} X_i^{\otimes 2t}, \tilde{\mathbb{E}} v^{\otimes 2t} \right\rangle \geqslant 10^t a^{2t}$ and*

$$\sum_{i \in S_g} \left( \frac{1}{n} - w_i \right) \leqslant \sum_{i \in S_b} \left( \frac{1}{n} - w_i \right).$$

*Then*

$$\sum_{i \in S_g} \left( \frac{1}{n} - w_i' \right) < \sum_{i \in S_b} \left( \frac{1}{n} - w_i' \right).$$

*Proof of Lemma C.3.* Note, that it is enough to show that

$$\sum_{i \in S_g} w_i - w_i' < \sum_{i \in S_b} w_i - w_i'.$$

Further, recall that $w_i' = \left( 1 - \frac{\tau_i}{\tau_{\max}} \right) w_i$, so for all $i \in [n]$, $w_i - w_i' = \frac{1}{\tau_{\max}} \tau_i w_i$. Hence is enough to show that

$$\sum_{i \in S_g} \tau_i w_i < \sum_{i \in S_b} \tau_i w_i.$$

Since $S_g$ and $S_b$ partition $[n]$ and

$$\sum_{i=1}^{n} w_i \tau_i = \left\langle \sum_{i=1}^{n} w_i X_i^{\otimes 2t}, \tilde{\mathbb{E}} v^{\otimes 2t} \right\rangle.$$

we can prove $\sum_{i \in S_g} \tau_i w_i < \sum_{i \in S_b} \tau_i w_i$ by showing that

$$\sum_{i \in S_g} \tau_i w_i \leqslant a^{2t} < \frac{\left\langle \sum_{i=1}^{n} w_i X_i^{\otimes t}, \tilde{\mathbb{E}} v^{\otimes 2t} \right\rangle}{2}.$$

Note that

$$\sum_{i \in S_g} \tau_i w_i = \left\langle \sum_{i \in S_g} w_i X_i^{\otimes 2t}, \tilde{\mathbb{E}} v^{\otimes 2t} \right\rangle \leqslant \left\langle \frac{1}{n} \sum_{i=1}^{n} \left( X_i^* \right)^{\otimes 2t}, \tilde{\mathbb{E}} v^{\otimes 2t} \right\rangle \leqslant a^{2t}.$$

$\square$

*Proof of Lemma C.2.* We will show that the algorithm terminates after at most $\lceil 2\varepsilon n \rceil$ iterations. Assume that it does not terminate after $T = \lceil 2\varepsilon n \rceil$ iterations. Note that the number of entries of $w$ that are equal to 0 increases by at least 1 in every iteration. Hence, after $T$ iterations we have set

at least $\varepsilon n$ entries of $w$ to zero whose index lies in $S_g$. By assumption that the algorithm did not terminate and Lemma C.3, it holds that

$$\varepsilon \leqslant \sum_{i \in S_g} \left( \frac{1}{n} - w_i^{(T)} \right) < \sum_{i \in S_b} \left( \frac{1}{n} - w_i^{(T)} \right) \leqslant \frac{|S_b|}{n} \leqslant \varepsilon \,,$$

a contradiction.

Let $T$ be the index of the last iteration of the algorithm before termination. Then

$$\left\| \frac{1}{n} - w^{(T)} \right\|_1 = \sum_{i \in S_g} \frac{1}{n} - w_i^{(T)} + \sum_{i \in S_b} \frac{1}{n} - w_i^{(T)} < 2 \sum_{i \in S_b} \frac{1}{n} - w_i^{(T)} \leqslant 2\varepsilon \,.$$

$\square$

# D  Sum-of-Squares Certificates

We use the standard sum-of-squares machinery, used in numerous prior works, e.g. [KS17a, KS17b, HL18, HL19, dKNS20, DKK$^+$22].

Let $f_1, f_2, \ldots, f_r$ and $g$ be multivariate polynomials in $x$. A *sum-of-squares proof* that the constraints $\{f_1 \geqslant 0, \ldots, f_m \geqslant 0\}$ imply the constraint $\{g \geqslant 0\}$ consists of sum-of-squares polynomials $(p_S)_{S \subseteq [m]}$ such that

$$g = \sum_{S \subseteq [m]} p_S \cdot \Pi_{i \in S} f_i \,.$$

We say that this proof has *degree* $\ell$ if for every set $S \subseteq [m]$, the polynomial $p_S \Pi_{i \in S} f_i$ has degree at most $\ell$. If there is a degree $\ell$ SoS proof that $\{f_i \geqslant 0 \mid i \leqslant r\}$ implies $\{g \geqslant 0\}$, we write:

$$\{f_i \geqslant 0 \mid i \leqslant r\} \,\big|\!\overline{{}_\ell} \, \{g \geqslant 0\} \,.$$

We provide degree $2\ell$ sum-of-squares proofs from the system $\mathcal{A}_K$ (see below) of $(n + d)^{O(1)}$ constraints. The sum-of-squares algorithm (that appeared it [Sho87, Par00, Nes00, Las01]. See, e.g., Theorem 2.6. [DKK$^+$22] for the precise formulation) returns a linear functional $\tilde{\mathbb{E}} : \mathbb{R}[x]_{\leqslant 2\ell} \to \mathbb{R}$, that is called a *degree $2\ell$ pseudo-expectation*, that satisfies the constraints of $\mathcal{A}_K$ in time $(n + d)^{O(\ell)}$. In particular, it means that once we prove in sum-of-squares of degree $2\ell$ that constraints $\mathcal{A}_K$ imply that some polynomial $g(u)$ is non-negative, the value of the $\tilde{\mathbb{E}}$ returned by the algorithm on $g(u)$ is also non-negative.

Recall the system $\mathcal{A}_K$ of elastic constraints in Equation (C.1) as follows:

$$\mathcal{A}_K : \begin{cases} \forall i \in [d] & s_i^2 = 1 \\ \forall i \in [d] & s_i v_i \geqslant v_i \\ \forall i \in [d] & s_i v_i \geqslant -v_i \\ & \displaystyle\sum_{i=1}^d v_i^2 \leqslant 1 \\ & \displaystyle\sum_{i=1}^d s_i v_i \leqslant \sqrt{K} \end{cases}$$

Also recall that the vectors from the elastic ball $\left\{ v \in \mathbb{R}^d \,\middle|\, \|v\| \leqslant 1 \,, \, \|v\|_1 \leqslant \sqrt{K} \right\}$ satisfy these constraints with $s_i = \text{sign}(v_i)$.

The following lemma is similar to Lemma 3.4 from [DKK$^+$22], but we prove it in using the elastic constraints. The derivation from the elastic constraints requires a bit more work.

**Lemma D.1.** *For arbitrary polynomial $p(v) = \sum_{1 \leqslant i_1, \ldots, i_t \leqslant d} p_{i_1 \ldots i_t} \cdot v_{i_1} \cdots v_{i_t}$ of degree at most $t$ we have*

$$\mathcal{A}_K \,\big|\!\overline{{}_{4t}^{s,v}} \, \left\{ (p(v))^2 \leqslant \|p\|_\infty^2 \cdot K^t \right\} \,.$$

*Proof.* Observe that $\mathcal{A}_K \left|\frac{s,v}{2}\right. s_i v_i \geqslant 0$, hence $\mathcal{A}_K \left|\frac{s,v}{4t}\right. \left(\sum_{i=1}^d s_i v_i\right)^{2t} \leqslant K^t$. In addition, by note that, $v_{i_1} \cdots v_{i_t} \leqslant s_{i_1} v_{i_1} \cdots s_{i_t} v_{i_t}$. It follows that

$$\mathcal{A}_K \left|\frac{s,v}{2t}\right. \left\{ \sum_{1 \leqslant i_1,\dots,i_t \leqslant d} p_{i_1\dots i_t} \cdot v_{i_1} \cdots v_{i_t} \leqslant \sum_{1 \leqslant i_1,\dots,i_t \leqslant d} |p_{i_1,\dots i_t}| s_{i_1} v_{i_1} \cdots s_{i_t} v_{i_t} \right\}$$

$$\left|\frac{s,v}{2t}\right. \left\{ - \sum_{1 \leqslant i_1,\dots,i_t \leqslant d} p_{i_1\dots i_t} \cdot v_{i_1} \cdots v_{i_t} \leqslant \sum_{1 \leqslant i_1,\dots,i_t \leqslant d} |p_{i_1,\dots i_t}| s_{i_1} v_{i_1} \cdots s_{i_t} v_{i_t} \right\}$$

$$\left|\frac{s,v}{4t}\right. \left\{ \left( \sum_{1 \leqslant i_1,\dots,i_t \leqslant d} p_{i_1\dots i_t} \cdot v_{i_1} \cdots v_{i_t} \right)^2 \leqslant \left( \sum_{1 \leqslant i_1,\dots,i_t \leqslant d} |p_{i_1,\dots i_t}| s_{i_1} v_{i_1} \cdots s_{i_t} v_{i_t} \right)^2 \right\}$$

$$\left|\frac{s,v}{4t}\right. \left\{ \left( \sum_{1 \leqslant i_1,\dots,i_t \leqslant d} p_{i_1\dots i_t} \cdot v_{i_1} \cdots v_{i_t} \right)^2 \leqslant \|p\|_\infty^2 \left( \sum_{i=1}^d s_i v_i \right)^{2t} \right\}$$

$$\left|\frac{s,v}{4t}\right. \left\{ \left( \sum_{1 \leqslant i_1,\dots,i_t \leqslant d} p_{i_1\dots i_t} \cdot v_{i_1} \cdots v_{i_t} \right)^2 \leqslant \|p\|_\infty^2 \cdot K^t \right\}.$$

$\square$

The following lemma shows that we can certify an upper bound on the value of the empirical moments (as polylinear functions) of truncated distribution $Z_i(\tau)$ on the vectors from the elastic ball.

**Lemma D.2** (Certifiable bound on empirical moments). *Suppose that for some $t, \ell \in \mathbb{N}$ and $M_{2t} \geqslant 1$,*

$$\mathcal{A}_K \left|\frac{s,v}{\ell}\right. \left\{ \mathbb{E}\langle X_1^*, v\rangle^{2t} \leqslant M_{2t}^{2t} \cdot \|\Sigma\|^t \right\},$$

*and for some $\nu \geqslant 1$*

$$\max_{j \in [d]} \mathbb{E}|X_{1j}^*|^{4t} \leqslant \nu^{4t} \cdot \|\Sigma\|^{2t}.$$

*If $\tau \gtrsim \nu^2 \cdot \sqrt{K} \cdot \sqrt{\|\Sigma\|}$ and*

$$n \geqslant 1000 \left( \nu^{4t} \cdot K^t + \left( \frac{\tau}{\sqrt{\|\Sigma\|}} \right)^{2t} \right) \cdot K^t \cdot t \log(d/\delta),$$

*then with probability at least $1 - \delta$, for each degree $2\ell$ pseudo-expectation $\tilde{\mathbb{E}}$ that satisfies $\mathcal{A}_K$,*

$$\tilde{\mathbb{E}} \left[ \frac{1}{n} \sum_{i=1}^n \langle X_i'(\tau), v\rangle^{2t} \right] \leqslant (2M_{2t})^{2t} \cdot \|\Sigma\|^t.$$

*Proof.* Consider the polynomial

$$p(v) = \frac{1}{n} \sum_{i=1}^n \langle X_i'(\tau), v\rangle^{2t} - \mathbb{E}\langle X_1^*, v\rangle^{2t},.$$

By Lemma E.6 and the assumptions on $n$ and $\tau$, its coefficients are bounded by

$$\Delta = 20 \sqrt{\frac{\nu^{4t} \|\Sigma\|^{2t} \cdot t \log(d/\delta)}{n}} + 20 \frac{\tau^{2t} \cdot t \log(d/\delta)}{n} + \frac{2t\nu^{4t} \cdot \|\Sigma\|^{2t}}{\tau^{2t}} \leqslant \frac{2^t M_{2t}^{2t} \cdot \|\Sigma\|^t}{K^t}.$$

It follows that

$$\mathcal{A}_K \left|\frac{s,v}{2\ell}\right. \left\{ \left( \frac{1}{n} \sum_{i=1}^n \langle X_i'(\tau), v\rangle^{2t} \right)^2 \leqslant \left( \frac{1}{n} \sum_{i=1}^n \langle X_i'(\tau), v\rangle^{2t} - \mathbb{E}\langle X_1^*, v\rangle^{2t} + \mathbb{E}\langle X_1^*, v\rangle^{2t} \right)^2 \right\}$$

$$\left| \frac{s,v}{2\ell} \left\{ \left( \frac{1}{n} \sum_{i=1}^{n} \langle X_i'(\tau), v \rangle^{2t} \right)^2 \leqslant 2 \left( \frac{1}{n} \sum_{i=1}^{n} \langle X_i'(\tau), v \rangle^{2t} - \mathbb{E} \langle X_1^*, v \rangle^{2t} \right)^2 + 2 \left( \mathbb{E} \langle X_1^*, v \rangle^{2t} \right)^2 \right\} \right.$$

$$\left| \frac{s,v}{2\ell} \left\{ \left( \frac{1}{n} \sum_{i=1}^{n} \langle X_i'(\tau), v \rangle^{2t} \right)^2 \leqslant 2\Delta^2 K^{2t} + 2M_{2t}^{4t} \|\Sigma\|^{2t} \right\} \right.$$

Hence

$$\tilde{\mathbb{E}} \left( \frac{1}{n} \sum_{i=1}^{n} \langle X_i, v \rangle^{2t} \right)^2 \leqslant (2M_{2t})^{4t} \cdot \|\Sigma\|^{2t} .$$

By Cauchy-Schwarz inequality for pseudo-expectations (see, for example, Fact A.2. from [DKK$^+$22]) we get the desired bound. □

# E  Properties of the truncation

As before, let $X_1^*, \ldots, X_n^*$ be iid samples from $\mathcal{D}$.

For $\tau > 0$, let $X_{ij}'(\tau) = X_{ij}^* \mathbf{1}_{\left[ |X_{ij}^*| \leqslant \tau \right]}$. In this section we prove some properties of $X_{ij}'(\tau)$ that we use in the paper. We start with the following lemma.

**Lemma E.1.** *Suppose that for some $\nu \geqslant 1$,*

$$\max_{j \in [d]} \mathbb{E} |X_{ij}^*|^4 \leqslant \nu^4 \cdot \|\Sigma\|^2 .$$

*Then*

$$\left\| \mathbb{E} \left( X_i'(\tau) - X_i^* \right) \left( X_i'(\tau) - X_i^* \right)^\top \right\|_\infty \leqslant \frac{\nu^4 \cdot \|\Sigma\|^2}{\tau^2} .$$

*Proof.*

$$\left| \mathbb{E} \left( X_{ij}'(\tau) - X_{ij}^* \right) \left( X_{ij'}'(\tau) - X_{ij'}^* \right) \right| = \left| \mathbb{E} X_{ij}^* \mathbf{1}_{\left[ |X_{ij}^*| > \tau \right]} X_{ij'}^* \mathbf{1}_{\left[ |X_{ij'}^*| > \tau \right]} \right|$$

$$\leqslant \sqrt{\mathbb{E} \mathbf{1}_{\left[ |X_{ij}^*| > \tau \right]} \left( X_{ij}^* \right)^2} \cdot \sqrt{\mathbb{E} \mathbf{1}_{\left[ |X_{ij'}^*| > \tau \right]} \left( X_{ij'}^* \right)^2}$$

$$\leqslant \left( \mathbb{E} \mathbf{1}_{\left[ |X_{ij}^*| > \tau \right]} \cdot \mathbb{E} \left( X_{ij}^* \right)^4 \right)^{1/4} \cdot \left( \mathbb{E} \mathbf{1}_{\left[ |X_{ij'}^*| > \tau \right]} \cdot \mathbb{E} \left( X_{ij'}^* \right)^4 \right)^{1/4}$$

$$\leqslant \left( \mathbb{P} \left[ |X_{ij}^*|^4 > \tau^4 \right] \cdot \mathbb{P} \left[ |X_{ij'}^*|^4 > \tau^4 \right] \right)^{1/4} \cdot \nu^2 \|\Sigma\|$$

$$\leqslant \nu^4 \|\Sigma\|^2 / \tau^2 .$$

□

The following lemma shows that the moments of the truncated distribution are close to the moments of $X_i^*$ in $\ell_\infty$-norm.

**Lemma E.2.** *Let $t \in \mathbb{N}$ and suppose that for some $B > 0$ and $q > 0$,*

$$\max_{j \in [d]} \mathbb{E} |X_{ij}^*|^{t+q} \leqslant B^{t+q} .$$

*Then*

$$\left\| \mathbb{E} \left( X_i'(\tau) \right)^{\otimes t} - \mathbb{E} \left( X_i^* \right)^{\otimes t} \right\|_\infty \leqslant \frac{t \cdot B^{t+q}}{\tau^q} .$$

*Proof.* Denote $a = X_i'(\tau)$, $b = X_i^*$. Note that by Hölder's inequality, for all $s \in [t]$,

$$\mathbb{E}|b_{j_1} \cdots b_{j_{s-1}}| \cdot |a_{j_s} - b_{j_s}| \cdot |a_{j_{s+1}} \cdots a_{j_t}| = \mathbb{E}|b_{j_1} \cdots b_{j_{s-1}} b_{j_s} a_{j_{s+1}} \cdots a_{j_t}| \cdot \mathbf{1}_{[a_{j_s}=0]}$$

$$\leqslant \left(\mathbb{P}\left[a_{j_s} = 0\right]\right)^{\frac{q}{t+q}} \cdot \left(\mathbb{E}|b_{j_1} \cdots b_{j_{s-1}} b_{j_s} a_{j_{s+1}} \cdots a_{j_t}|^{1+q/t}\right)^{\frac{t}{t+q}}$$

$$\leqslant \left(\mathbb{P}\left[(X_{ij_s}^*)^{t+q} > \tau^{t+q}\right]\right)^{\frac{q}{t+q}} \cdot \left(\mathbb{E}|b_{j_1} \cdots b_{j_t}|^{1+q/t}\right)^{\frac{t}{t+q}}$$

$$\leqslant \frac{B^q}{\tau^q} \cdot \left(\max_{j \in [d]} \mathbb{E}|b_j|^{t+q}\right)^{\frac{t}{t+q}}$$

$$\leqslant \frac{B^{t+q}}{\tau^q}$$

It follows that

$$\left|\mathbb{E}\, a_{j_1} a_{j_2} \cdots a_{j_t} - \mathbb{E}\, b_{j_1} b_{j_2} \cdots b_{j_t}\right| \leqslant \mathbb{E}\left|a_{j_1} a_{j_2} \cdots a_{j_t} - b_{j_1} b_{j_2} \cdots b_{j_t}\right|$$

$$\leqslant \mathbb{E}\left|a_{j_1} a_{j_2} \cdots a_{j_t} - b_{j_1} a_{j_2} \cdots a_{j_t} + b_{j_1} a_{j_2} \cdots a_{j_t} - b_{j_1} b_{j_2} \cdots b_{j_t}\right|$$

$$\leqslant \mathbb{E}|a_{j_1} - b_{j_1}| \cdot |a_{j_2} \cdots a_{j_t}| + \mathbb{E}|b_{j_1}| \cdot |a_{j_2} \cdots a_{j_t} - b_{j_2} \cdots b_{j_t}|$$

$$\leqslant \frac{t \cdot B^{t+q}}{\tau^q} \,.$$

$\square$

The following statement is a straightforward corollary of Lemma E.2 with $q = t$:

**Corollary E.3.** *Let $t \in \mathbb{N}$ and suppose that for some $B > 0$,*

$$\max_{j \in [d]} \mathbb{E}|X_{ij}^*|^{2t} \leqslant B^{2t} \,.$$

*Then*

$$\left\|\mathbb{E}\left(X_i'(\tau)\right)^{\otimes t} - \mathbb{E}\left(X_i^*\right)^{\otimes t}\right\|_\infty \leqslant \frac{t \cdot B^{2t}}{\tau^t} \,.$$

The following two statements are special cases of Corollary E.3 for $t = 1$ and $t = 2$.

**Corollary E.4.**

$$\left\|\mathbb{E}\, X_i'(\tau)\right\|_\infty \leqslant \frac{\|\Sigma\|}{\tau} \,.$$

**Corollary E.5.** *Suppose that for some $v \geqslant 1$,*

$$\max_{j \in [d]} \mathbb{E}|X_{ij}^*|^4 \leqslant v^4 \cdot \|\Sigma\|^2 \,.$$

*Then*

$$\left\|\mathbb{E}\left(X_i'(\tau)\right)\left(X_i'(\tau)\right)^\top - \mathbb{E}\left(X_i^*\right)\left(X_i^*\right)^\top\right\|_\infty \leqslant \frac{2 \cdot v^4 \cdot \|\Sigma\|^2}{\tau^2} \,.$$

The following lemma shows that the empirical mean of $\left(X_i'(\tau)\right)^{\otimes t}$ is close to $\mathbb{E}\left(X_1^*\right)^{\otimes t}$ for an appropriate choice of $\tau$ and large enough $n$.

**Lemma E.6.** *Let $t \in \mathbb{N}$ be and suppose that for some $v \geqslant 1$*

$$\max_{j \in [d]} \mathbb{E}|X_{ij}^*|^{2t} \leqslant v^{2t} \cdot \|\Sigma\|^t \,.$$

*Then with probability $1 - \delta$,*

$$\left\|\frac{1}{n}\sum_{i=1}^n \left(X_i'(\tau)\right)^{\otimes t} - \mathbb{E}\left(X_1^*\right)^{\otimes t}\right\|_\infty \leqslant 10\sqrt{\frac{v^{2t} \cdot \|\Sigma\|^t \cdot t \log(d/\delta)}{n}} + 10\frac{\tau^t \cdot t \log(d/\delta)}{n} + \frac{t \cdot v^{2t} \cdot \|\Sigma\|^t}{\tau^t} \,.$$

*Proof.* It follows from Corollary E.3, Bernstein inequality Fact I.1, and a union bound over all $d^t$ entries of $\mathbb{E}\left(X_1^*\right)^{\otimes t}$. $\square$

# F    Properties of sparse vectors

**Lemma F.1.** *Let $\Sigma \in \mathbb{R}^{d \times d}$ be a positive definite matrix, $k', k'' \in \mathbb{N}$, $r, \delta \geqslant 0$, and*

$$\mathcal{E}_{k'}(r) = \left\{ u \in \mathbb{R}^d \ \middle|\ \|\Sigma^{1/2}u\| \leqslant r\,,\ \|u\|_1 \leqslant \sqrt{k'} \cdot r \right\},$$

$$\mathcal{S}_{k''}(r) = \left\{ u \in \mathbb{R}^d \ \middle|\ \|\Sigma^{1/2}u\| = (1+\delta) \cdot r\,,\ u \text{ is } k''\text{-sparse} \right\}.$$

*If $k'' \geqslant 4k'\|\Sigma\|/\delta^2$, then*

$$\mathcal{E}_{k'}(r) \subseteq \operatorname{conv}(\mathcal{S}_{k''}(r)).$$

*Proof.* Let us take some $u \in \mathcal{E}_{k'}(r)$. Without loss of generality assume that $u_1 \geqslant u_2 \geqslant \ldots \geqslant u_d$. Let's split indices $\{1, 2, \ldots, d\}$ into blocks $B_1 \ldots, B_{\lceil d/k'' \rceil}$ of size $k''$ (the last block might be of smaller size). Let for each block $B_i$, let

$$p_i = \frac{\|\Sigma^{1/2}u_{B_i}\|}{\sum_{j=1}^{\lceil d/k'' \rceil} \|\Sigma^{1/2}u_{B_j}\|}$$

Since $\sum_i^{\lceil d/k'' \rceil} p_i = 1$ and $u = \sum_i^{\lceil d/k'' \rceil} p_i u_{B_i}/p_i$, it is sufficient to show that for all $i$, $\|\Sigma^{1/2}u_{B_i}\|/p_i \leqslant (1+\delta)r$.

Note that for all $j \geqslant 2$, since $\|u_{B_j}\| \leqslant \sqrt{k''}\|u_{B_j}\|_\infty$ and $\|u_{B_j}\|_\infty \leqslant \frac{1}{k''}\|u_{B_{j-1}}\|_1$,

$$\|\Sigma^{1/2}u_{B_j}\| \leqslant \sqrt{\|\Sigma\|} \cdot \|u_{B_j}\| \leqslant \sqrt{k''\|\Sigma\|} \cdot \|u_{B_j}\|_\infty \leqslant \sqrt{\frac{\|\Sigma\|}{k''}} \cdot \|u_{B_{j-1}}\|_1 \,.$$

By the triangle inequality,

$$\|\Sigma^{1/2}u_{B_1}\| \leqslant \|\Sigma^{1/2}u\| + \sum_{j=2}^{\lceil d/k'' \rceil} \|\Sigma^{1/2}u_{B_j}\|\,.$$

Hence

$$
\begin{aligned}
\frac{\|\Sigma^{1/2}u_{B_i}\|}{p_i} &= \sum_{j=1}^{\lceil d/k'' \rceil} \|\Sigma^{1/2}u_{B_j}\| \\
&\leqslant \|\Sigma^{1/2}u\| + 2\sum_{j=2}^{\lceil d/k'' \rceil} \|\Sigma^{1/2}u_{B_j}\| \\
&\leqslant r + 2\sqrt{\frac{\|\Sigma\|}{k''}} \sum_{j=2}^{\lceil d/k'' \rceil} \|u_{B_{j-1}}\|_1 \\
&\leqslant r + 2\sqrt{\frac{\|\Sigma\|}{k''}} \|u\|_1 \\
&\leqslant \left(1 + 2\sqrt{\frac{k'\|\Sigma\|}{k''}}\right) \cdot r \\
&\leqslant (1+\delta) \cdot r\,.
\end{aligned}
$$

$\square$

**Lemma F.2.** *Let $\Sigma \in \mathbb{R}^{d \times d}$ be a positive definite matrix, and let $X \in \mathbb{R}^{m \times d}$ be a matrix such that for some $r > 0$ and $\delta \in (0, 1)$, for all $k''$-sparse vectors $u'$ such that $r \leqslant \|\Sigma^{1/2}u'\| \leqslant 2r$,*

$$(1 - \delta) \cdot \|\Sigma^{1/2}u'\| \leqslant \tfrac{1}{\sqrt{m}}\|Xu'\| \leqslant (1+\delta) \cdot \|\Sigma^{1/2}u'\|$$

*If $k'' \geqslant 4k'\|\Sigma\|/\delta^2$, then for all $u$ such that $\|\Sigma^{1/2}u\| = r$ and $\|u\|_1 \leqslant r\sqrt{k'}$,*

$$(1 - 4\delta) \cdot r \leqslant \tfrac{1}{\sqrt{m}}\|Xu\| \leqslant (1+4\delta) \cdot r\,.$$

*Proof.* The inequality $\frac{1}{\sqrt{m}}\|Xu\| \leq (1 + \delta)^2 \cdot r \leq (1 + 4\delta) \cdot r$ follows from Jensen's inequality and Lemma F.1.

Let us show that $(1 - 4\delta) \cdot r \leq \frac{1}{\sqrt{m}}\|Xu\|$. Let $B_1, \ldots, B_{\lceil d/k''\rceil}$ be blocks of indices as in the proof of Lemma F.1. It follows that

$$\frac{1}{\sqrt{m}}\|Xu\| \geq \frac{1}{\sqrt{m}}\|Xu_{B_1}\| - \sum_{j=2}^{\lceil d/k''\rceil} \frac{1}{\sqrt{m}}\|Xu_{B_j}\|$$

$$\geq (1 - \delta) \cdot \|\Sigma^{1/2}u_{B_1}\| - (1 + \delta) \sum_{j=2}^{\lceil d/k''\rceil} \|\Sigma^{1/2}u_{B_j}\|$$

$$\geq (1 - \delta) \cdot \|\Sigma^{1/2}u\| - 2 \cdot r\sqrt{\frac{k'\|\Sigma\|}{k''}}$$

$$\geq (1 - \delta)^2 \cdot r - \delta r$$

$$\geq (1 - 4\delta) \cdot r \,.$$

$\square$

# G  Lower bounds

In this section we prove Statistical Query lower bounds. SQ lower bounds is a standard tool of showing computational lower bounds for statistical estimation and decision problems. SQ algorithms do not use samples, but have access to an oracle that can return the expectation of any bounded function (up to a desired additive error, called *tolerance*). The SQ lower bounds formally show the trateoff between the number of queries to the oracle and the tolerance. The standard interpretation of SQ lower bounds relies on the fact that simulating a query with small tolerance using iid samples requires large number of samples. Hence these lower bounds are interpreted as a tradeoff between the time complexity (number of queries) and sample complexity (tolerance) of estimators. See [DKS17] for more details.

First we give necessary definitions. These definitions are standard and can be found in [DKS17].

**Definition G.1** (STAT Oracle)**.** Let $\mathcal{D}$ be a distribution over $\mathbb{R}^d$. A *statistical query* is a function $f : \mathbb{R}^d \to [-1, 1]$. For $\tau > 0$ the STAT($\tau$) oracle responds to the query $f$ with a value $v$ such that $|v - \mathbb{E}_{X \sim \mathcal{D}} f(X)| \leq \tau$. Parameter $\tau$ is called the *tolerance* of the statistical query.

Simulating a query STAT($\tau$) normally requires $\Omega(1/\tau^2)$ iid samples from $\mathcal{D}$, hence SQ lower bounds provide a trade-off between the running time (number of queries) and the sample complexity ($\Omega(1/\tau^2)$).

**Definition G.2** (Pairwise Correlation)**.** Let $\mathcal{D}_1, \mathcal{D}_2, \mathcal{D}$ be absolutely continuous distributions over $\mathbb{R}^d$, and suppose that $\text{supp}(\mathcal{D}) = \mathbb{R}^d$. The *pairwise correlation* of $\mathcal{D}_1$ and $\mathcal{D}_2$ with respect to $\mathcal{D}$ is defined as

$$\chi_D(\mathcal{D}_1, \mathcal{D}_2) = \int_{\mathbb{R}^d} \frac{p_{\mathcal{D}_1}(x)p_{\mathcal{D}_2}(x)}{p_{\mathcal{D}}(x)}dx - 1\,,$$

where $p_{\mathcal{D}_1}(x), p_{\mathcal{D}_2}(x), p_{\mathcal{D}}(x)$ are densities of $\mathcal{D}_1, \mathcal{D}_2, \mathcal{D}$ respectively.

**Definition G.3** (Chi-Squared Divergence)**.** Let $\mathcal{D}', \mathcal{D}$ be absolutely continuous distributions over $\mathbb{R}^d$, and suppose that $\text{supp}(\mathcal{D}) = \mathbb{R}^d$. The *chi-squared divergence* from $\mathcal{D}'$ to $\mathcal{D}$ is

$$\chi^2(\mathcal{D}', \mathcal{D}) = \chi_D(\mathcal{D}', \mathcal{D}')\,.$$

**Definition G.4** (($\gamma, \rho$)-correlation)**.** Let $\rho, \gamma > 0$, and let $\mathcal{D}$ be a distribution over $\mathbb{R}^d$. We say that a family of distributions $\mathcal{F}$ over $\mathbb{R}^d$ is *($\gamma, \rho$)-correlated relative to $\mathcal{D}$*, if for all distinct $\mathcal{D}', \mathcal{D}'' \in \mathcal{F}$, $|\chi_D(\mathcal{D}', \mathcal{D}'')| \leq \gamma$ and $|\chi_D(\mathcal{D}', \mathcal{D}')| \leq \rho$.

**Fact G.5.** *Let $\mathcal{D}$ be a distribution over $\mathbb{R}^d$ and $\mathcal{F}$ be a family of distributions over $\mathbb{R}^d$ that does not contain $\mathcal{D}$, and consider a hypothesis testing problem of determining whether a given distribution $\mathcal{D}' = \mathcal{D}$ or $\mathcal{D}' \in \mathcal{F}$.*

Let $\gamma, \rho > 0$, $s \in \mathbb{N}$, and suppose that there exists a subfamily of $\mathcal{F}$ of size $s$ that is $(\gamma, \rho)$-correlated relative to $\mathcal{D}$. Then for all $\gamma' > 0$, every SQ algorithm for the hypothesis testing problem requires queries of tolerance $\sqrt{\gamma + \gamma'}$ or makes at least $s\gamma'/(\rho - \gamma)$ queries.

We will also need the following facts:

**Fact G.6** ([DKS17, Lemma 6.7]). *Let $c \in (0, 1)$ and $k, d \in \mathbb{N}$ be such that $k \leqslant \sqrt{d}$. There exists a set $\mathcal{V} \subset \mathbb{R}^d$ of $k$-sparse unit vectors of size $d^{ck^c/8}$ such that for all distinct $u, v \in \mathcal{V}$, $\langle v, u \rangle \leqslant 2k^{c-1}$.*

**Fact G.7** ([DKS17, Lemma 3.4]). *Let $m \in \mathbb{N}$, and suppose that a distribution $\mathcal{M}$ over $\mathbb{R}$ matches first $m$ moments of $N(0, 1)$. For a unit vector $v \in \mathbb{R}^d$ let $\mathcal{P}_v$ be a distribution such that its projections onto the direction of $v$ has distribution $\mathcal{M}$, the projection onto the orthogonal complement is $N(0, \mathrm{Id}_{d-1})$, and these projections are independent. Then for all $u, v \in \mathbb{R}^d$,*

$$\left| \chi_{N(0,\mathrm{Id}_d)}(\mathcal{P}_v, \mathcal{P}_u) \right| \leqslant |\langle u, v \rangle|^{m+1} \chi^2(\mathcal{M}, N(0, 1)).$$

The following fact is a slight reformulation of Lemma E.4 from [DKS19]

**Fact G.8** ([DKS19, Lemma E.4]). *Let $y \sim N(0, 1)$, $\mu_0 > 0$, $m \in \mathbb{N}$, and $g : \mathbb{R} \to \mathbb{R}$. Let $\mathcal{M}_\mu$ be a family of distributions over $\mathbb{R}$ satisfies the following properties:*

1. $\mathcal{M} = (1 - \varepsilon_\mu)N(\mu, \Theta(1)) + \varepsilon_\mu \mathcal{B}_\mu$ *for some $\varepsilon_\mu$ and $\mathcal{B}_\mu$ such that $\mathcal{M}_\mu$ has the same first $m$ moments as $N(0, 1)$.*

2. *If $|\mu| \geqslant 10\mu_0$, then $\varepsilon_\mu/(1 - \varepsilon_\mu) \leqslant O(\mu^2)$ and $\chi^2(\mathcal{M}, N(0, 1)) \leqslant e^{O(\max\{1/\mu^2, \mu^2\})}$.*

3. *If $|\mu| \leqslant 10\mu_0$, then $\varepsilon_\mu = \varepsilon$ and $\chi^2(\mathcal{M}, N(0, 1)) \leqslant g(\varepsilon)$.*

*For unit $v \in \mathbb{R}^d$ let $\mathcal{P}_{v,\mu}$ be the same as $\mathcal{P}_v$ in Fact G.7 whose projection onto $v$ is $\mathcal{M}_\mu$. Let $Q'_v$ be a distribution over $\mathbb{R}^{d+1}$ such that $(X, y) \sim Q'_v$ satisfy the following properties: $y \sim N(0, 1)$, and $X|y \sim \mathcal{P}_{v, \mu_0 \cdot y}$. Then for all unit $u, v \in \mathbb{R}^d$,*

$$\chi_{\mathcal{D}}(Q'_v, Q'_u) \leqslant (g(\varepsilon) + O(1)) \cdot |\langle v, u \rangle|^{m+1},$$

*where $\mathcal{D} = N(0, \mathrm{Id}_{d+1})$.*

**Proposition G.9** (Formal version of Proposition 1.10). *Let $k, d \in \mathbb{N}$, $k \leqslant \sqrt{d}$, $\varepsilon \in (0, 1/2)$, $c \in (0, 1)$. For a vector $\beta^* \in \mathbb{R}^d$, and a number $\sigma > 0$, consider the distribution $\mathcal{G}(\beta^*, \sigma)$ over $\mathbb{R}^{d+1}$ such that $(X, y) \sim \mathcal{G}(\beta^*, \sigma)$ satisfy $X \sim N(0, \mathrm{Id})$ and $y = \langle X, \beta^* \rangle + \eta$, where $\eta \sim N(0, \sigma^2)$ is independent of $X$.*

*There exist a set $\mathcal{B} \subset \mathbb{R}^d$ of $k$-sparse vectors, $0.99 \leqslant \sigma \leqslant 1$ and a distribution $Q$ over $\mathbb{R}^{d+1}$, such that if an SQ algorithm $\mathcal{A}$ given access to a mixture $(1 - \varepsilon)\mathcal{G}(\beta^*, \Sigma, \sigma) + \varepsilon Q$ for $\beta^* \in \mathcal{B}$, outputs $\hat{\beta}^*$ such that $\|\beta^* - \hat{\beta}\| \leqslant 10^{-5}$, then $\mathcal{A}$ either*

- *makes $d^{ck^c/8} \cdot k^{-2+2c}$ queries,*

- *or makes at least one query with tolerance smaller than $k^{-1+c}e^{O(1/\varepsilon^2)}$.*

*Proof.* Note that $(X, y) \sim \mathcal{G}(\beta^*, \sigma)$ satisfy $y \sim N(0, \sigma_y^2)$, where $\sigma_y^2 = \|\beta^*\|^2 + \sigma^2$ and $X|y \sim N\left(\frac{y}{\sigma_y^2}\beta^*, \mathrm{Id} - \frac{1}{\sigma_y^2}\beta^*\beta^{*\top}\right)$.

We will use vectors $\beta^*$ of norm $10^{-5}$. Denote $v = \beta^*/\|\beta^*\|$ and let $\sigma^2 = 1 - \|\beta^*\|^2$. Consider a distribution $\mathcal{M}_\mu = (1 - \varepsilon)N\left(\mu, 1 - \|\beta^*\|^2\right) + \varepsilon N\left(-\frac{1-\varepsilon}{\varepsilon}\mu, 1\right)$. Note that $\chi^2\left(\mathcal{M}_\mu, N(0, 1)\right) \leqslant e^{O(\max\{1/\mu^2, \mu^2\})}$, and $\varepsilon/(1 - \varepsilon) \leqslant O(\mu^2)$ for $\mu \geqslant 10^{-4}$. Hence by Fact G.8,

$$\chi_{\mathcal{D}}(Q'_v, Q'_u) \leqslant e^{O(1/\varepsilon^2)}\langle v, u \rangle^2$$

for all unit $v, u \in \mathbb{R}^d$.

Using Fact G.6, we can apply Fact G.5 with $\gamma = k^{2c-2}e^{O(1/\varepsilon^2)}$, $\rho = e^{O(1/\varepsilon^2)}$, $\gamma' = (\rho - \gamma) \cdot k^{-2+2c}$, we get that $\mathcal{A}$ requires at least $d^{ck^c/8}k^{-2+2c}$ queries with tolerance greater than $k^{-1+c}e^{O(1/\varepsilon^2)}$. $\quad\square$

**Proposition G.10** (Formal version of Proposition 1.11). *Let $k, d \in \mathbb{N}$, $k \leqslant \sqrt{d}$, $\varepsilon \in (0, 1/2)$, $c \in (0, 1)$. For a vector $\beta^* \in \mathbb{R}^d$, a positive definite matrix $\Sigma$ and a number $\sigma > 0$, consider the distribution $\mathcal{G}(\beta^*, \Sigma, \sigma)$ over $\mathbb{R}^{d+1}$ such that $(X, y) \sim \mathcal{G}(\beta^*, \Sigma, \sigma)$ satisfy $X \sim N(0, \Sigma)$ and $y = \langle X, \beta^* \rangle + \eta$, where $\eta \sim N(0, \sigma^2)$ is independent of $X$.*

*There exist a set $\mathcal{B} \subset \mathbb{R}^d$ of $k$-sparse vectors, $\frac{1}{2}\mathrm{Id} \preceq \Sigma \preceq \mathrm{Id}$, $0.99 \leqslant \sigma \leqslant 1$ and a distribution $\mathcal{Q}$ over $\mathbb{R}^{d+1}$, such that if an SQ algorithm $\mathcal{A}$ given access to a mixture $(1 - \varepsilon)\mathcal{G}(\beta^*, \Sigma, \sigma) + \varepsilon\mathcal{Q}$ for $\beta^* \in \mathcal{B}$, outputs $\hat{\beta}$ such that $\|\beta^* - \hat{\beta}\| \leqslant 10^{-5}\sqrt{\varepsilon}$, then $\mathcal{A}$ either*

- *makes $d^{ck^c/8} \cdot k^{-4+4c}$ queries,*

- *or makes at least one query with tolerance at least $k^{-2+2c}e^{O(1/\varepsilon)}$.*

*Proof.* Note that $(X, y) \sim \mathcal{G}(\beta^*, \Sigma, \sigma)$ satisfy $y \sim N(0, \sigma_y^2)$, where $\sigma_y^2 = \beta^{*\top}\Sigma\beta^* + \sigma^2$ and $X|y \sim N\left(\frac{y}{\sigma_y^2}\Sigma\beta^*, \Sigma - \frac{1}{\sigma_y^2}(\Sigma\beta^*)(\Sigma\beta^*)^\top\right)$.

We will use vectors $\beta^*$ of norm $10^{-5}\sqrt{\varepsilon}$. Denote $v = \beta^*/\|\beta^*\|$ and let $\sigma^2 = 1 - \beta^{*\top}\Sigma\beta^*$, $\Sigma = \mathrm{Id} - c'vv^\top$, where $c'$ is a constant such that

$$\Sigma - \frac{1}{\sigma_y^2}(\Sigma\beta^*)(\Sigma\beta^*)^\top = \mathrm{Id} - c'vv^\top - \left(10^{-5}(1 - c')^2\varepsilon\right)vv^\top = \mathrm{Id} - vv^\top/3.$$

By [DKS19, Lemmas E.2], there exists a distribution $\mathcal{M}$ that satisfies the assumption of Fact G.8 with $m = 3$ and $g(\varepsilon) = e^{O(1/\varepsilon)}$. Hence by Fact G.8,

$$\chi_{\mathcal{D}}(\mathcal{Q}_v', \mathcal{Q}_u') \leqslant e^{O(1/\varepsilon)}\langle v, u \rangle^4$$

for all unit $v, u \in \mathbb{R}^d$.

Using Fact G.6, we can apply Fact G.5 with $\gamma = k^{4c-4}e^{O(1/\varepsilon)}$, $\rho = e^{O(1/\varepsilon)}$, $\gamma' = (\rho - \gamma) \cdot k^{-4+4c}$, we get that $\mathcal{A}$ requires at least $d^{ck^c/8}k^{-4+4c}$ queries with tolerance smaller than $k^{-2+2c}e^{O(1/\varepsilon)}$. $\square$

## H   Sub-exponential designs

Recall that a distribution $\mathcal{D}$ in $\mathbb{R}^d$ is called $L$-sub-exponential, if it has $(Lt)$-bounded $t$-th moment for each $t \in \mathbb{N}$. In particular, all log-concave distributions are $L$-sub-exponential for some $L \leqslant O(1)$.

In this section we discuss how we can improve the dependence of the sample complexity on $\varepsilon$ if (in addition to the assumptions of Theorem B.3) we assume that $\mathcal{D}$ is $L$-sub-exponential. For these designs we do not need a truncation.

First, let us show how the gradient bound Lemma B.5 modifies in this case. It can be obtained directly from Bernstein's inequality for sub-exponential distributions ([RH23, Theorem 1.13])

**Lemma H.1.** *With probability at least $1 - \delta/10$,*

$$\left\|\frac{1}{n}\sum_{i \in [n]}\phi(\eta_i)X_i^*\right\|_\infty \leqslant 10\sqrt{\frac{\|\Sigma\|\log(d/\delta)}{n}} + 10\frac{\sqrt{\|\Sigma\|} \cdot L \cdot \log(d/\delta)}{n}.$$

The proof of strong convexity bound (Lemma B.6) is exactly the same, with $X'(\tau) = X^*$.

Finally, we need to bound $\left\|\frac{1}{n}\sum_{i=1}^n (X_i^*)^{\otimes 2t} - \mathbb{E}(X_1^*)^{\otimes 2t}\right\|_\infty$, since we need to prove Lemma D.2 for sub-exponential distributions. By Lemma C.1. from [DKK+22], for all $L$-sub-exponential distributions, with probability $1 - \delta$,

$$\left\|\frac{1}{n}\sum_{i=1}^n (X_i^*)^{\otimes 2t} - \mathbb{E}(X_1^*)^{\otimes 2t}\right\|_\infty \leqslant O\left(\sqrt{\frac{t\log(d/\delta)}{n}} \cdot \left(10L\sqrt{\|\Sigma\|} \cdot t^2\log(d/\delta)\right)^{2t}\right)$$

Hence with $n \gtrsim K^{2t} \cdot \left(10t^2 \log(d/\delta)\right)^{4t+1}$, we get $\left\| \frac{1}{n} \sum_{i=1}^{n} \left(X_i^*\right)^{\otimes 2t} - \mathbb{E}\left(X_1^*\right)^{\otimes 2t} \right\|_\infty \leqslant L^{2t} \|\Sigma\|^t / K^t$, so we get the conclusion of Lemma D.2.

Putting everything together, the sample complexity is

$$n \gtrsim \frac{k \log(d/\delta)}{\varepsilon^{2-1/t}} + \frac{\left(k^2 \log(d/\delta)\right) 10^{5s/(s-2)} M_s^{s/(s-2)} \kappa(\Sigma)^{4+s/(s-2)}}{\alpha} + k^{2t} \cdot \kappa(\Sigma)^{2t} \cdot \left(10^6 \cdot t^2 \log(d/\delta)\right)^{4t+1}.$$

Note that we can use $s = 4$ and $M_s = 4L$.

Consider the case when $\mathcal{D}$ is log-concave, so $L \leqslant O(1)$. For $\kappa(\Sigma) \leqslant O(1)$, $\alpha \geqslant \Omega(1)$, $t = 1$, with high probability we get error $O(\sigma\sqrt{\varepsilon})$ as long as

$$n \gtrsim \frac{k \log d}{\varepsilon} + k^2 \cdot (\log d)^5.$$

Similarly, for $\kappa(\Sigma) \leqslant O(1)$, $\alpha \geqslant \Omega(1)$, $t = 2$, with high probability we get error $O(M\sigma\varepsilon^{3/4})$ (where $M \leqslant O(\sqrt{\log d})$ is the same as in Theorem 1.7) as long as

$$n \gtrsim \frac{k \log d}{\varepsilon^{3/2}} + k^4 \cdot (\log d)^9.$$

Note that for sub-Gaussian distributions one can use better tail bounds and the polylog($d$) factors should be better in this case.

# I  Concentration Bounds

Throughout the paper we use the following versions of versions of Bernstein's inequality. The proofs can be found in [Tro15].

**Fact I.1** (Bernstein inequality). *Let $L > 0$ and let $x \in \mathbb{R}^d$ be a zero-mean random variable. Let $x_1, \ldots, x_n$ be i.i.d. copies of $x$. Suppose that $|x| \leqslant L$. Then the estimator $\bar{x} = \frac{1}{n} \sum_{i=1}^{n} x_i$ satisfies for all $t > 0$*

$$\mathbb{P}(|\bar{x}| \geqslant t) \leqslant 2 \cdot \exp\left(-\frac{t^2 n}{2\mathbb{E}x^2 + Lt}\right).$$

**Fact I.2** (Bernstein inequality for covariance). *Let $L > 0$ and let $x \in \mathbb{R}^d$ be a d-dimensional random vector. Let $x_1, \ldots, x_n$ be i.i.d. copies of $x$. Suppose that $\|x\|^2 \leqslant L$. Then the estimator $\bar{\Sigma} = \frac{1}{n} \sum_{i=1}^{n} x_i x_i^\top$ satisfies for all $t > 0$*

$$\mathbb{P}\left(\left\|\bar{\Sigma} - \mathbb{E}xx^\top\right\| \geqslant t\right) \leqslant 2d \cdot \exp\left(-\frac{t^2 n}{2L\|\Sigma\| + Lt}\right).$$

