# OpenReview forum: "Robust Sparse Regression with Non-Isotropic Designs"
_NeurIPS.cc/2024/Conference — NeurIPS 2024 poster_

### Official Review · Reviewer_3A8o · 2024-07-07

**Soundness:** 3
**Presentation:** 3
**Contribution:** 3
**Rating:** 6
**Confidence:** 2

**Summary:**

The authors provide a computationally efficient estimator for robust and sparse linear regression under non-isotropic covariance matrices. Their first result achieves $O(\sqrt{\epsilon})$ error with state-of-the-art sample complexity under a weaker noise assumption than prior work. Their second result is the first to achieve $o(\sqrt{\epsilon})$ error for robust and sparse linear regression under non-isotropic covariance matrices, under a sum-of-squares certificate for the $4^{th}$ moment. They further provide Statistica Query Lower bounds supporting their results.

**Strengths:**

- They improve the state-of-the-art result for $O(\sqrt{\epsilon})$ error in terms of the sample complexity being independent of the $l-2$ norm of the true weight vector, $\beta_{*}$
- They provide the first result achieving $o(\sqrt{\epsilon})$ error, under a suitable sum-of-squares certificate for the $4^{th}$ moment.
- They provide SQ lower bounds which suggest that their results might be tight for polynomial time estimators

**Weaknesses:**

I don't see any weaknesses per se, was just curious to understand if the authors believe that the sum-of-squares assumption is necessary to achieve $o(\sqrt{\epsilon})$ error or a weaker assumption such as Definition 1.6 suffices.

**Questions:**

Small typo on Line 110 - ehtr

See weakness section.

---

> ### Author Rebuttal · Authors · 2024-08-07
>
> Dear Reviewer 3A8o,
>
> Thank you very much for your review! We appreciate your evaluation of our work.
>
> Regarding your question: We believe that moment assumptions without sum-of-squares are not enough.  For robust mean estimation, similar lower bound was shown in [HL19]. However, we believe that it might be possible that some additional assumptions, in particular, related to symmetry of the distribution, might be potentially used without any assumptions related to sum-of-squares, as was shown for robust mean estimation in [NST23].
>
> References:
>
> [HL19] *How Hard is Robust Mean Estimation?* Samuel B. Hopkins, Jerry Li. **COLT 2019**
>
> [NST23] *Robust Mean Estimation Without Moments for Symmetric Distributions*. Gleb Novikov, David Steurer, Stefan Tiegel. **NeurIPS 2023**

---

### Official Review · Reviewer_e1Aa · 2024-07-11

**Soundness:** 2
**Presentation:** 1
**Contribution:** 3
**Rating:** 5
**Confidence:** 2

**Summary:**

This paper studies sparse linear regression $y^* = X^* \beta^* + \eta$ (where $\beta^*$ is $k$-sparse) in the presence of two types of adversaries.

- First, the noise vector $\eta$ is sampled before $X^*$ in an (obliviously) adversarial way. Then $X^*$ is sampled independently of $\eta$ with i.i.d. rows and $y^*$ is generated according to the model.

- Second, an adaptive adversary takes $(X^*, y^*)$ and (adaptively) corrupts at most $\epsilon$ fraction of $(X_i^*, y_i^*)$.
The resulting samples given to the statistician are denoted by $(X, y)$.

The results are two-fold.

- First, a poly-time algorithm for estimating $\beta^*$ up to $\epsilon$ expected prediction error using $n \approx k^2/\epsilon^2$ samples.

- Second, a poly-time algorithm for estimating $\beta^*$ up to $o(\epsilon)$ expected prediction error using $n \approx k^4/\epsilon^6$ samples.
For correlated Gaussian design, this can be improved to $n\approx k^4/\epsilon^4$.

**Strengths:**

NA

**Weaknesses:**

NA

**Questions:**

I'm not familiar with the SoS machinery, so the authors may find my comments shallow and unconstructive.
That said, I am familiar with ordinary (sparse) linear regression and I do think that the presentation of the paper should be improved.

Major comments:

1. The sample complexity bounds are written in a wacky way.
Why don't we fix the goal to be achieving $\epsilon$ error and write the sample complexity bounds accordingly?

1. The two results in the paper (for achieving $O(\sqrt{\epsilon})$ and $o(\sqrt{\epsilon})$ errors) seem to have very discontinuous transitions as $\epsilon$ varies.
Can the authors comment on whether it's expected to be so or it's a proof artifact?
In fact, should I think of $\epsilon$ as potentially depending on $n,k,d$ in an arbitrary way?
This is not clear since sometimes the authors view $\epsilon$ as a constant in which case $O(\epsilon), o(\epsilon)$ don't make sense to me.

1. I suggest the authors add a short section discussing related work at the level of techniques (e.g., filtering, SoS, etc.).
Currently, these techniques and the difference between the present paper and prior work are buried in Section 2 which itself is written in an obscure way.

Minor comments:

1. Line 48-51 seems out of place.

1. Please unify the notation $N(0,\mathrm{Id}), N(0,1)^n$.

1. Multiple typos in line 110.

1. Line 114, should $X$ be $X^*$?

1. Typo in line 234.

1. I don't understand line 243-244.
Please define "simulation".
My shallow understanding is that SQ lower bounds only apply to detection version of the problem.
I'm not sure if this has the same spirit as "simulation".

1. Section 2 (Techniques) is written in a very informal and dense way.
As someone who's not familiar with these techniques, I couldn't get much out of it.

1. Typo in line 259, $\eta\sim N(0,\mathrm{Id})$.

1. Typo in line 264, it --> in.

1. Line 291, please provide a reference for the sentence in the parentheses.

1. Is line 298 the only property one needs from the output of filtering?
If so, why not say it upfront in line 266?

1. Line 364, 365, there're two "only"'s in the sentence.

1. I don't understand line 371, even grammatically.

1. A very minor comment on the writing. I feel that the language used in this paper is quite informal and the writing was perhaps done in a hasty.
For instance, there're ~20 "only", ~20 "even", ~10 "likely"...

---

> ### Author Rebuttal · Authors · 2024-08-06
>
> Dear Reviewer e1Aa
>
> Thank you very much for your review! We will address all the typos and enhance the writing and notation as per your suggestions.
>
> 1) *“The sample complexity bounds are written in a wacky way. Why don't we fix the goal to be achieving $\varepsilon$ error and write the sample complexity bounds accordingly?”* — For some design distributions it might be not possible achieve error $\varepsilon$, even information-theoretically. Furthermore, even for Gaussian designs error $O(\varepsilon)$ is believed to be not achievable by polynomial time computable estimators. Hence it makes sense to ask the question what error can we get in polynomial time, and what are the weakest assumptions needed to achieve this error.
>
> 2) *“The two results in the paper (for achieving $O(\sqrt{\varepsilon})$  and $o(\sqrt{\varepsilon})$ errors) seem to have very discontinuous transitions as $\varepsilon$ varies. Can the authors comment on whether it's expected to be so or it's a proof artifact? In fact, should I think of  as potentially depending on $n,k,d$ in an arbitrary way? This is not clear since sometimes the authors view $\varepsilon$ as a constant in which case $O(\varepsilon)$, $o(\varepsilon)$ don't make sense to me.”* — indeed, the transition is discontinuous (we need $k^2$ samples for $O(\varepsilon)$ and $k^4$ samples for $o(\varepsilon)$), and the lower bound (Proposition  11) shows that it is likely to be necessary for polynomial time algorithms. While $\varepsilon$ can depend arbitrarily on $n,k,d$ (as long as the conditions of the theorems are satisfied), it is instructive to think of $\varepsilon$ as something tending to zero as $k\to\infty$, but not very rapidly, e.g. $1/polylog(k)$ or so. The reason is that for Gaussian designs there is a simple algorithm that achieves error $O(\varepsilon \sqrt{k})$, which is better than our bounds for very small $\varepsilon$, but is much worse for larger $\varepsilon$. It is a usual phenomenon in robust statistics, and the standard goal is to obtain a bound that would not (directly) depend on the dimension, but would only depend on $\varepsilon$ (note that a term that depends on $\varepsilon$ and does not vanish for fixed $\varepsilon$ as $n\to \infty$ is inevitable). And while the results are correct for small constant $\varepsilon$, they are interesting only under assumption $\varepsilon \to 0$. We will add a more detailed discussion on this topic to the final version of the paper.
>
> 3) We will improve Section 2 by simplifying technical details and emphasizing the difference from the techniques of the prior works. The main reason why our Theorem 1.4 is better than the result of the prior work ([Sas22]) is that we use a different proof of the strong convexity (we use eps-net over sparse vectors, and then extend the argument to the elastic vectors, while they used a different argument).
>
> 4) Regarding SQ lower bounds: in general, SQ algorithms can be used for estimation problems, not only for hypothesis testing problems. However, for lower bounds we indeed use standard arguments that SQ algorithms need either very small tolerance, or very large number of queries in order to solve some hypothesis testing problem (and that implies that SQ algorithms with small number of queries and large tolerance cannot estimate $\beta^*$ up to certain error). By "simulation" we simply mean the standard simulation of the oracle using iid samples, i.e. estimating $\mathbb{E} f(X)$ using the empirical mean. We will clarify it in the final version of the paper. We discuss SQ lower bounds (and some definitions related to SQ algorithms) in Appendix G. See also [DKS17] for more details regarding SQ algorithms and lower bounds.
>
> 5) Regarding line 114: Indeed, there is a typo, there should be $X^*$, not $X$.
>
>
> 6) Regarding line 371: We simply meant that the set of indices $A$ is independent of $X^*$, and hence we may assume that the rows of $X^*$ that correspond to $A$ are just iid samples from $D$. We will clarify it in the camera version of the paper.
>
>
> Thank you again for your valuable feedback, which we will address to improve the final version of our paper.

---

> ### Comment · Reviewer_e1Aa · 2024-08-08
>
> I thank the reviewer for the detailed response.
> In particular, regarding item 1 and 2 in the response:
>
> 1. I realize that my previous question here didn't make much sense. Sorry about that and thanks anyway for the clarification.
>
> 2. Thanks for the discussion on the scaling of $\epsilon$ which is helpful and, as the authors mentioned, should be included in the paper somewhere.
>
> My overall evaluation remains unchanged.

---

### Official Review · Reviewer_mVbA · 2024-07-12

**Soundness:** 2
**Presentation:** 3
**Contribution:** 3
**Rating:** 5
**Confidence:** 2

**Summary:**

It developed efficient estimators for sparse linear regression in the presence of oblivious and adaptive adversaries. It presents a robust algorithm that outperforms sota under desired conditions on the moment of the distribution.

**Strengths:**

- The paper introduces several robust algorithms that outperform the sota in sparse linear regression.

- It also gives analysis of weighted penalized Huber loss suitable for heavy-tailed designs.

- The paper includes theoretical results with proofs and assumptions, which are clearly stated.

**Weaknesses:**

- algorithms may require significant computational resources, which could limit practical applications.

- There is a lack of empirical evidence comparison with the sota even on a toy setup.

**Questions:**

NA.

**Limitations:**

The results depend on specific assumptions about the distribution and moments of the data, which may not always hold in real-world scenarios.

---

> ### Author Rebuttal · Authors · 2024-08-07
>
> Dear Reviewer mVbA,
>
> Thank you very much for your review! Our focus was on proving the strongest possible theoretical guarantees in polynomial time. There is extensive literature in learning theory, particularly in robust estimation and linear regression, that lacks empirical validation, and we believe making these theoretical results practical is a significant challenge in machine learning and theoretical computer science. Addressing this challenge is beyond the scope of our work.
>
> We identify two issues for practical applications: First, the algorithms for the $o(\sqrt{\varepsilon})$ error require $k^4$ samples, which might be too large for some instances. Second, our approach uses the sum-of-squares algorithm, which is very slow in practice. While the first issue is likely to be inherent, the second can be potentially addressed in future works. Specifically, it may be possible to design a fast spectral algorithm inspired by the sum-of-squares approach (although this is beyond our current scope).

---

> > ### Comment · Reviewer_mVbA · 2024-08-13
> >
> > I thank the authors for their response. My scores are unchanged.

---

### Official Review · Reviewer_sWxz · 2024-07-12

**Soundness:** 3
**Presentation:** 1
**Contribution:** 2
**Rating:** 4
**Confidence:** 3

**Summary:**

The paper focuses on robust estimation of sparse linear regression in the presence of both oblivious and adaptive adversaries. It claims to offer polynomial-time algorithms capable of recovering a sparse coefficient vector with high probability. The paper includes theoretical analyses using the standard techniques such as pre-filtering, truncation to ensure robustness.

**Strengths:**

- The authors claim to deal with two adversaries, which can be seen as a generalization of dealing with one adversary.

- The theoretical analysis seems to be rigorous.

- The analysis also covers heavy tails under the standard assumptions on the moments made in the existing literature.

- The literature review is good.

**Weaknesses:**

As per my understanding, the oblivious and adaptive adversary are dealt in literature separately.

- For the oblivious adversary, the existing literature can handle constant outlier proportions and deliver consistent estimates.

- For adaptive adversary, existing literature [PFL20] already has the results which are proposed in the submission. They use a trimmed MLE algorithm whose convergence rates are analyzed. Further, [Sas22] and [SF23] already extended these results for sparse settings (line 264).

So, as per my understanding, the authors claim bounds similar to those in the existing literature for a slightly more complex setting of two adversaries.

- Why are few entries of $\eta$ required to be close to $1$ in magnitude? Doesn't that make the oblivious adversary weak?

- The authors use pre-filtering ideas, assuming a known covariance matrix and truncation to deal with heavy tails. They use standard techniques like sum-of-squares and weighted Huber loss minimization with $\ell_1$ regularization.

These ideas are well explored in the literature. So, strictly speaking, there is a lack of major novelty in new ideas to deal with outliers and encourage robustness.

- More importantly, $O(\sqrt{\varepsilon})$ error is already proved to be minimax optimal in the existing literature even when $n \to \infty$ while using techniques like trimming, pre-filtering, or truncation. A more interesting question could be under what conditions on $\varepsilon \neq 0$ or adversary can we achieve zero error for at least $n \to \infty$.

**Questions:**

- Is the oblivious adversary allowed to change all the $n$ samples, unlike the adaptive adversary, which only changes $\varepsilon*n$ samples?

- If the oblivious adversary is also restricted to $\varepsilon * n$ samples, it should be stated in line 6. Also, are the oblivious and adaptive adversaries changing the same $\varepsilon * n$ samples?

- The authors claim that their algorithm achieves $o(\sqrt{\varepsilon})$ error for all log-concave distributions if $\varepsilon \leq 1/\text{polylog}(d)$. Hence, $\varepsilon$ can be very small for large $d$. Can this upper bound on $\varepsilon$ be improved?

- The condition number of the covariance matrix is assumed to be bounded. Is that assumed in the existing literature or a new assumption in this submission?

- Line 110 typo: 'ehrtr' and from 's' more

- Line 234 typo: 'ghdsample' complexity

**Limitations:**

- In Definition 2.1, why was $|x_i| \leq 2$ chosen as the threshold to define the quadratic and linear part of Huber loss? In general, it could be dependent on the data? For example, there could be some dataset where $|x_i| \leq 3$ may be more appropriate.

- The above could affect the upper bound on the gradient $|\phi(\eta)|$ taken as 2 in line 288.

- A conclusion section and pointers to future work seem to be missing.

- Towards the end of the main manuscript, the focus seems too much on the proof sketch. The writing or presentation can be improved.

-  Some sort of empirical justification for the proposed results on synthetic or real datasets would have been really nice. This would have also helped to draw attention to the significance of the contributions of polynomial time algorithms. In Proposition 1.10 and 1.11, the number of queries is claimed as $\exp(d^{\Omega(1)})$, which does not seem to be very useful for practitioners.

---

> ### Author Rebuttal · Authors · 2024-08-06
>
> Dear Reviewer sWxz,
>
> Thank you very much for your review! We appreciate your feedback and would like to address the points raised, as well as answer the questions and discuss limitations.
>
> **Regarding the weaknesses:**
>
> *Existing literature on the adaptive adversary:* While [PJL20] inspired the idea of filtering before Huber loss minimization, our work diverges significantly. [PFL20] did not explore the sparse case and (in the case of $o(\sqrt{\varepsilon})$ error) studied only isotropic designs. Our approach employs a sum-of-squares program for sparse regression, which was not addressed in prior works. Consequently, our Theorem 1.7 presents a smaller error than prior works on sparse regression ([Sas22, SF23]). Even in Theorem 1.4's context, dealing with very heavy-tailed designs without sum-of-squares, our results surpass [Sas22] with weaker moment assumptions. We remark that our results outperform the results of prior works even in the simplest special case of Gaussian noise (i.e. when Gaussian noise is added to $X^*\beta^*$ by an oblivious adversary).
>
> *Oblivious adversary's strength:* The assumption that a few entries of $\eta$ are close to 1 appeared in prior works on linear regression with oblivious adversary ([dNS21, dLN+21]). This is one of the weakest assumptions used in literature (so the adversary is strong). Without this assumption, the oblivious adversary can simply add Gaussian noise with very large variance (say, $2^d$) to each entry of $X^*\beta^*$, and the error has to scale with the variance of the noise. Hence some restriction of this kind is necessary for achieving optimal (or at least bounded) error.
>
> *Covariance matrix assumption:* We do not assume that the covariance matrix is known. Furthermore, the problem with unknown covariance that we study is likely to be significantly more complicated than the case when the covariance is known.
>
> *Minimax optimality of $O(\sqrt{\varepsilon})$ error:* While generally optimal, specific designs can yield better errors. Our Theorem 1.7 is novel in demonstrating errors smaller than $O(\sqrt{\varepsilon})$ in sparse settings, outperforming prior results, even for Gaussian designs and Gaussian noise (i.e. when the oblivious adversary simply adds iid Gaussian variables to the entries of $X^* \beta^*$).
>
> *Zero error conditions:* Achieving zero error is impossible even with $\varepsilon = 0$ when Gaussian noise is added to $X^*\beta^*$. If the query concerns settings without $n$-independent error terms, our focus on the classical version of the adaptive adversary necessitates some error term that depends on $\varepsilon$ but not (directly) on $n$. Weaker adaptive adversary versions are beyond our scope.
>
> **Regarding the questions:**
>
> *Oblivious adversary's power:* Yes, the oblivious adversary can alter all samples, provided it remains independent of $X^*$. A constant fraction (0.01 in Definition 1.1) of true labels can be changed by at most $\sigma$, while other samples can be changed by arbitrarily large values. This model includes adding Gaussian or Cauchy noise, aligning with previous noise models (see also the discussion in lines 120--125, where we explain how the general random noise model studied in prior works is captured by the oblivious adversary).
>
> *$\varepsilon < 1/polylog(d)$ requirement:* This stems from the KLS conjecture. If the conjecture is true, this assumption is needed. For some specific log-concave distributions (e.g. Gaussian distribution, or uniform distributions over $\ell_p$ balls) for which Poincare parameter is know to be independent of $d$, this assumption on $\varepsilon$ is not needed. We discuss it in lines 203--219, and we will clarify this further in the final version of the paper.
>
> *Error dependence on covariance condition number:* The error's dependence on the covariance condition number is detailed in Theorem B.3, and it is similar to prior works ([Sas22], [SF23]). We did not optimize this dependence, but Proposition 1.11 highlights that even a condition number of 2 presents significant challenges compared to the case of identity covariance.
>
> **Regarding the limitations:**
>
> 1) *On the Huber parameter*: In Definition 2.1, we focus on $\sigma = 1$, using Huber parameter 2. For general cases, dividing by $\sigma$ suffices, as this parameter is assumed to be known. Note that in prior works on regression with oblivious adversary ([SBRJ19, dNS21, dLN+21]) this parameter is also assumed to be known, see discussion in [SBRJ19] on "Estimating the noise variance". We will add a discussion on this topic to the final version of the paper.
>
> 2) *On the lower bounds*: Our focus was on proving the strongest possible theoretical guarantees in polynomial time, and we added the lower bounds in order to illustrate that our assumptions on the number of samples are inherent for polynomial time algorithms. Our results are consistent with the usage of SQ lower bounds in prior works on learning theory, e.g. [DKS17]. We believe that practical applications of theoretical results on algorithmic robust statistics are a significant challenge in machine learning and theoretical computer science. Addressing this challenge is beyond the scope of our work.
>
> Thank you again for your valuable feedback. We hope this addresses your concerns and clarifies our contributions.

---

> > ### Comment · Reviewer_sWxz · 2024-08-12
> >
> > I thank the authors for their detailed response. I will keep my original score.

---

### Official Review · Reviewer_JPFs · 2024-07-22

**Soundness:** 4
**Presentation:** 4
**Contribution:** 4
**Rating:** 7
**Confidence:** 3

**Summary:**

This paper studies sparse linear regression with oblivious and adaptive adversaries. The design matrix is random, the noise is chosen by an adversary, and the goal is to find the optimal k-sparse weight vector. The results are as follows: to achieve $O(\sqrt{\varepsilon})$ error, a sample complexity of $\widetilde{O}(k^2/\varepsilon)}$ is needed, and to achieve $O(\varepsilon^{3/4})$ error, a sample complexity of $\widetilde{O}(k^4/\varepsilon^3)$ is needed. Previous work did not achieve error better than $\sqrt{\varepsilon}$ with any sample complexity better than $d^2$, unless the rows of a design matrix are drawn from an isotropic Gaussian distribution.

To obtain their results, the authors make the following assumptions: (1) the noise vector has at least $0.01n$ entries that are bounded by $\sigma$, where $n$ is the number of examples, (2) at most an $\varepsilon$ fraction of rows of $X$/entries of $y$ are corrupted, and (3) the rows of $X$ have bounded 3rd moments and entrywise-bounded 4th moments, i.e. the rows of X can be heavy-tailed random vectors.

In this setting, they obtain two results:
1. With $O(k^2/\varepsilon)$ samples, they obtain $O(\sqrt{\varepsilon})$ error. Compared to the best prior work for heavy-tailed design matrices, the advantage of this result is that there is no dependence on the norm of the optimal k-sparse regression vector.
2. With $k^4/\varepsilon^3$ samples, they obtain $O(\varepsilon^{3/4})$ error. This result additionally assumes that the distribution of the rows of the design matrix has a bounded 4th moment, with a constant-degree sum-of-squares proof of the bound. It also additionally assumes that the rows of the design matrix have entrywise-bounded 8th moment.
Additionally, they give matching statistical query lower bounds.

The algorithm consists of obtaining a filtered set of examples, then minimizing Huber loss with l1 regularization, on the remaining examples. In the analysis, the authors show that if the following two conditions:
(1) A bound on the gradient of the loss function
(2) A condition similar to strong convexity
hold on the boundary of certain regions they call "elastic balls" (the elastic ball of radius $r$ has points with $\ell_2$ norm at most $r$ and $\ell_1$ norm at most $\sqrt{k} \cdot r$) then the weight vector they obtain will be similar to the true optimal weight vector.

One of the key steps in this analysis, compared to prior works is as follows: instead of using Cauchy-Schwarz to bound certain terms in the gradient bound, they use Holder's inequality. This leads to them having to bound the sum of $\langle X_i, u \rangle^4$, as $X_i$ ranges over the examples in the filtered set, and $u$ can be any vector in the elastic ball of radius 1. Thus, during the algorithm, they must select $\widehat{S}$ so that it satisfies this property, and thus need an efficient algorithm to find such a set of size $(1 - O(\varepsilon))n$. They do this using a sum-of-squares relaxation with elastic constraints, which makes it so the relaxation set is a superset of $u^{\otimes 4}$ for $u$ in the elastic ball.

For the heavy-tailed case, it is also necessary to do some thresholding of X, which may have large entries even without adversarial noise. In previous works, the thresholding parameter needs to depend polynomially on the norm of the weight vector (to preserve the relationship between the un-corrupted X and y). This paper observes that in order to preserve the relationship between X and y, it is only necessary to analyze the effect of thresholding for the entries that are in the support of the optimal k-sparse regression vector $\beta^*$. This allows the thresholding parameter to only depend on $k$ and $\varepsilon$ and not the norm of $\beta^*$.

The proof strategy for showing the strong convexity property for heavy-tailed design matrices is also interesting. They first show it for vectors $u$ in the elastic balls which are $k$-sparse, then extend to dense vectors $u$. They show the bound for all k-sparse vectors using Bernstein's inequality, and a small $\epsilon$-net for the k-sparse vectors. To extend to dense vectors, they use the fact that if a quadratic form is $\Theta(r^2)$ on $k$-sparse vectors of norm $r$, then the quadratic form is $\Theta(r^2)$ on the elastic ball of radius $r$.

**Strengths:**

- The results are very strong compared to previous works.
- The technical tools used are very interesting.

**Weaknesses:**

- While the writing is mostly good, it would be nice to organize the techniques section better. It would be useful for readers if you explicitly write the algorithm in a separate box, and divide the techniques section into some subsections.

**Questions:**

- On line 278, what is $\hat{\beta}$? I thought the final estimator you return is $\hat{\beta}_{\hat{S}}$. Do you mean to refer to the ground-truth $k$-sparse regression vector?

**Limitations:**

Yes.

---

> ### Author Rebuttal · Authors · 2024-08-07
>
> Dear Reviewer JPFs,
>
> Thank you very much for your review! We appreciate your evaluation of our work and will improve the techniques section in the final version according to your suggestions.
>
> Regarding your question: Indeed, it is a typo, there should be $\beta^*$ instead of $\hat{\beta}$ in line 278.

---

> > ### Comment · Reviewer_JPFs · 2024-08-13
> >
> > Thank you for your reply. I will keep my original score.

---

### Decision · Program_Chairs · 2024-09-25

**Decision:**

Accept (poster)

**Comment:**

This paper studies sparse linear regression robust against both oblivious and adaptive adversaries. The majority of reviewers agree that both the theoretical results and the technical contributions seem qualitatively strong. Although concerns about the practical nature of the proposed algorithms were raised, the general consensus is that the paper is primarily of theoretical interest. Author responses during the rebuttal phase adequately addressed additional initial reviewer concerns.

I recommend the authors incorporate reviewer feedback and improve the clarity/presentation to ensure the work has a stronger impact. Further, it has been brought to my attention two additional works that examine similar error models in related contexts: https://ieeexplore.ieee.org/document/4544953 and https://dl.acm.org/doi/10.1145/1250790.1250804. While these studies focus on problems where the signal is not required to be sparse, considering these connections may provide useful insights or strengthen the theoretical foundations of the current work. I encourage the authors to review these references and evaluate whether citing them could benefit their contribution.